# Differentiable Structure Learning with Ancestral Constraints

**Taiyu Ban** [1] **Changxin Rong** [1] **Xiangyu Wang** [1] **Lyuzhou Chen** [1] **Xin Wang** [1] **Derui Lyu** [1] **Qinrui Zhu** [1] **Huanhuan Chen** [1]

## Abstract

Differentiable structure learning of causal directed acyclic graphs (DAGs) is an emerging field in causal discovery, leveraging powerful neural learners. However, the incorporation of ancestral constraints, essential for representing abstract prior causal knowledge, remains an open research challenge. This paper addresses this gap by introducing a generalized framework for integrating ancestral constraints. Specifically, we identify two key issues: the non-equivalence of relaxed characterizations for representing path existence and order violations among paths during optimization. In response, we propose a binary-masked characterization method and an order-guided optimization strategy, tailored to address these challenges. We provide theoretical justification for the correctness of our approach, complemented by experimental evaluations on both synthetic and real-world datasets.

## 1. Introduction

Structure learning of causal DAGs is a fundamental task for uncovering causal relationships among variables from data (Pearl, 2000). Recent advances have reformulated this task as a continuous optimization problem by representing the discrete acyclicity constraint of the graph as a smooth equality, a technique known as differentiable structure learning (Zheng et al., 2018). This approach facilitates the use of powerful neural learners and gradient-based optimization methods to model complex causal mechanisms (Zheng et al., 2020; Yu et al., 2019), significantly advancing the fields of causal discovery and trustworthy AI (Zhang et al., 2021; Dai & Chen, 2022; Ruan et al., 2022; Wei et al., 2022).

In practice, it is common to have prior knowledge about the presence or absence of causal relationships, which must be incorporated into structure learning to accurately recover the underlying causal mechanisms (Constantinou et al., 2023; Amirkhani et al., 2016). When prior knowledge is abstract and does not directly identify a specific causal interaction in the data, ancestral constraints, which specify the existence or absence of either edges or indirect paths, provide a robust way to encode such abstract knowledge (Li & Beek, 2018).

For traditional structure learning methods based on score-and-search, ancestral constraints can be incorporated by verifying path presence conditions in a graph and eliminating graphs that violate the constraints (Chen et al., 2016), or by re-scoring graphs with additional bonuses for adherence to prior knowledge (O'Donnell et al., 2006).

In contrast, differentiable structure learning of causal DAGs relies on gradient-based optimization, which demands a smooth characterization that equivalently represents path presence conditions. Furthermore, the optimization process must be effectively guided toward favorable optima under continuous ancestral constraints, avoiding the risk of convergence to suboptimal points. These two distinctive requirements pose significant challenges to the integration of ancestral constraints in differentiable structure learning, leaving it an open research question in the field.

To address this research gap, we propose a generalized framework for integrating ancestral constraints into differentiable structure learning. First, we identify the issue of non-equivalence in continuous relaxed characterizations for representing path existence and propose a binary-masked characterization that accurately and equivalently represents path existence. Next, we address the order-violating issue among individual paths specified in the constraints, which can result in incomplete adherence to the constraints and lead to suboptimal optimization. To guide the optimization process toward a favorable optima, we propose an order-guided optimization strategy. It begins by enforcing partial order constraints implied by path existence, deriving a DAG that satisfies all specified path orders. The resulting order-consistent DAG is then used as the initial adjacency matrix to optimize the path existence-constrained problem.

We provide the necessary theoretical analysis and compre-

[1]School of Computer Science and Technology, University of Science and Technology of China, Hefei, China. Correspondence to: Xiangyu Wang <sa312@ustc.edu.cn>, Huanhuan Chen <hchen@ustc.edu.cn>.

*Proceedings of the 42nd International Conference on Machine Learning*, Vancouver, Canada. PMLR 267, 2025. Copyright 2025 by the author(s).

hensive empirical evidence on both synthetic and real-world datasets to validate our approach. Additionally, related works, proofs of statements, and complete experimental results are detailed in the appendix.

The main contributions are summarized as follows:

- To the best of our knowledge, this is the first paper to systematically address the integration of ancestral constraints into differentiable structure learning, enabling the use of abstract prior knowledge to guide the discovery of fine-grained causal mechanisms.

- We analyze the issue of non-equivalence in representing path existence using continuous relaxed characterizations and propose a binary-masked characterization that equivalently represents path existence.

- We identify the order-violation issue among paths, which can lead to suboptimal optimization, and introduce an order-guided optimization strategy that effectively guides the process toward favorable optima.

## 2. Preliminaries

This section introduces key concepts and notations, provides an overview of differentiable structure learning, and concludes with the connection between the smooth characterization of acyclicity and path absence.

### 2.1. Notations

Let $W \in \mathbb{R}^{d \times d}$ represent a weighted adjacency matrix, with the graph it defines denoted as $G(W)$. The node set of $G(W)$ is $X = \{x_i\}_{i=1}^{d}$, and the edge set is $E(G(W))$. The existence of a directed path from $x_i$ to $x_j$ in $G(W)$ is represented by $x_i \rightsquigarrow x_j$, while $x_i \overset{k}{\rightsquigarrow} x_j$ denotes a path of length $k$. The symbol $x_i$ and its index $i$ are used interchangeably to denote a node.

For matrix operations, $|W|$ denotes the element-wise absolute value, $W^k$ represents the $k$-th power of $W$, and $W_{i,j}^k$ refers to the $(i, j)$-th element of $W^k$. Operations such as $W - a$ or $aW$, where $a$ is a constant, are performed element-wise as subtraction or multiplication by $a$, respectively.

### 2.2. Differentiable Structure Learning

**Causal Graphical Model.** We start by defining the causal graphical model commonly used to model causal relationships among continuous variables.

**Definition 1** (Structural Equation Model (SEM)). An SEM is a directed acyclic graph (DAG) $G$ with a node set $X = \{x_i\}_{i=1}^{d}$, representing random variables, and a set of functions $F = \{f_i\}_{i=1}^{d}$, such that:

$$x_i = f_i(\text{pa}_i^G, z_i), \qquad (1)$$

where $\text{pa}_i^G$ denotes the parent node set of $x_i$ in $G$, and $z_i$ represents independent zero-mean noise terms $\{z_i\}_{i=1}^{d}$.

**Traditional Structure Learning.** Structure learning for an SEM $\langle G, F \rangle$ involves recovering the graph structure $G$ from a dataset $D \in \mathbb{R}^{d \times d}$ generated based on $G$ and $F$. Assume the SEM is parameterized by $\theta$, where the graph $G$ is represented by $W_\theta \in \mathbb{R}^{d \times d}$, with $(W_\theta)_{i,j} \neq 0$ indicating an edge $(x_i, x_j)$, and the functions $F$ are represented by $F_\theta$. The structure learning task can be formulated as:

$$\min_{\theta} \|F_\theta(D, W_\theta) - D\|_2^\ell + \lambda \|\theta\|_1,$$
$$\text{subject to } G(W_\theta) \in \text{DAG}. \qquad (2)$$

Here, $\| \cdot \|_2^\ell$ denotes the least-squares loss, $\| \cdot \|_1$ the $\ell_1$-norm, and $\lambda > 0$ the regularization weight. For simplicity, we focus on the linear case of structure learning, where the model parameters can be directly represented by a weighted adjacency matrix $W \in \mathbb{R}^{d \times d}$:

$$\min_{W \in \mathbb{R}^{d \times d}} \|DW - D\|_2^\ell + \lambda \|W\|_1,$$
$$\text{subject to } G(W) \in \text{DAG}. \qquad (3)$$

Since the acyclicity constraint $G(W) \in \text{DAG}$ is discrete, traditional methods rely on combinatorial optimization (Chickering, 2002) or other discrete approaches (Spirtes & Glymour, 1991).

**Smooth Characterization of Acyclicity.** To address the discrete nature of DAG structure learning, Zheng et al. introduced NOTEARS, which provides a smooth characterization of acyclicity via the trace exponential function:

$$h(W) \equiv \text{Trace}\left(e^{W \circ W}\right) - d = 0, \qquad (4)$$

where $\circ$ denotes the element-wise product, and $d$ is the node number. This characterization satisfies $h(W) = 0$ if and only if $G(W)$ is a DAG. Using this, the discrete optimization problem in Equation (3) is reformulated as a continuous optimization problem with an equality constraint:

$$\min_{W \in \mathbb{R}^{d \times d}} \|DW - D\|_2^\ell + \lambda \|W\|_1, \quad \text{s.t. } h(W) = 0. \quad (5)$$

**Optimization and Thresholding.** NOTEARS solves this problem using the augmented Lagrangian method, ensuring that $h(W)$ remains sufficiently small to enforce the acyclicity constraint. The graph structure $G(W)$ is then derived by applying a threshold to $|W|$. This framework has become foundational in the field and is widely adopted in subsequent work (Zheng et al., 2020; Sun et al., 2023).

### 2.3. Connection between Acyclicity and Path Absence

The function $h(W)$ representing the ayclicity degree of $G(W)$ in Equation (4) is constructed following the idea of

forbidding all paths from each node to itself (cycles):

$$G(W) \in \text{DAG} \iff \forall i, \neg (x_i \rightsquigarrow x_i \in G(W)). \quad (6)$$

To derive the smooth characterization of acyclicity, we first illustrate the characterization of path absence. Consider:

$$|W|_{i,j}^k = \sum_{q_1, q_2, \ldots, q_{k-1}} |W|_{i,q_1} |W|_{q_1,q_2} \cdots |W|_{q_{k-1},j}, \quad (7)$$

where $|W|_{i,j}^k$ represents the non-negative weighted sum of all length-$k$ paths from $x_i$ to $x_j$, with each path's weight being the product of the edge weights along that path. Thus:

$$|W|_{i,j}^k = 0 \iff \neg \left( x_i \overset{k}{\rightsquigarrow} x_j \in G(W) \right), \quad (8)$$

indicating the absence of a length-$k$ path between $x_i$ and $x_j$. Extending this to paths of any length up to $d$, we derive:

$$\sum_{k=1}^d |W|_{i,j}^k = 0 \iff \forall k < d, \neg \left( x_i \overset{k}{\rightsquigarrow} x_j \in G(W) \right) \\ \iff \neg (x_i \rightsquigarrow x_j \in G(W)). \quad (9)$$

The second equivalence follows from the contrapositive of the following proposition:

**Proposition 1.** *If there exists a directed path $(x_i, x_j)$ in a directed graph $G$ with $d$ nodes, then there exists at least one length-$k$ path $(x_i, x_j)$ in $G$ such that $k \leq d$.*

Equation (9) characterizes the absence of paths. On this basis, acyclicity in Equation (6), the absence of cycles, can equivalently be expressed as a smooth equality:

$$\forall i, \neg (x_i \rightsquigarrow x_i \in G(W)) \iff \sum_{k=1}^d \sum_{i=1}^d |W|_{i,i}^k = 0 \\ \iff \sum_{k=1}^d \text{Trace} \left( |W|^k \right) = 0. \quad (10)$$

From this, we derive the general form of the smooth characterization of acyclicity:

$$\hat{h}(W) \equiv \text{Trace} \left( \sum_{k=1}^d c_k |W|^k \right) = 0, \quad c_k > 0. \quad (11)$$

**Corollary 1.** *(Theorem 1 in (Wei et al., 2020)) A directed graph $G(W)$ is a DAG if and only if $\hat{h}(W) = 0$ for any $\hat{h}$ defined in Equation (11).*

The acyclicity function $h(W)$ of NOTEARS in Equation (4) is a special case of $\hat{h}(W)$, containing infinite terms to forbid cycles of all lengths while being computationally efficient for gradient-based optimization. Other acyclicity functions, such as $\text{Trace}((I + \frac{1}{d}B)^d - I)$ (Yu et al., 2019) and $-\log \det(sI - B) + d \log s$ (Bello et al., 2022), have also been proposed for efficient gradient calculation.

**Remark 1.** The above derivation provides an intuitive and accurate characterization of path absence in Equation (9), which underpins the construction of acyclicity characterizations central to differentiable structure learning. While path absence itself is not discussed further, it serves as a foundational concept for path existence characterization.

## 3. Continuous Characterization of Path Existence Constraints

This section begins by characterizing path existence through continuous relaxation. We then analyze its non-equivalent issue in representing path existences. To address this, we introduce a binary-masked continuous relaxation that provides an equivalent representation. Finally, we formulate the path existence-constrained task.

### 3.1. Continuous Relaxation of Path Existence

**Characterization via Continuous Relaxation.** We first recall the continuous characterization of path absence:

$$(p(W))_{i,j} = 0 \iff \neg (x_i \rightsquigarrow x_j \in G(W)), \\ \text{where } p(W) \equiv \sum_{k=1}^d |W|^k. \quad (12)$$

Since $p(W)$ is non-negative, its contrapositive equivalence provides a direct characterization of path existence:

$$(p(W))_{i,j} > 0 \iff x_i \rightsquigarrow x_j \in G(W), \quad (13)$$

which states that a nonzero value in $p(W)$ implies the existence of the corresponding directed path.

**Proposition 2.** *For a graph $G(W)$ where edge existence follows $W_{i,j} \neq 0 \iff (x_i, x_j) \in E(G(W))$, at least one directed path from $x_i$ to $x_j$ exists if and only if $(p(W))_{i,j} > 0$, where $p(W)$ is defined in Equation (12).*

However, in the continuous optimization of structure learning, edge weights $W_{i,j}$ are rarely exactly zero. Instead, a threshold $\epsilon_0 > 0$ is used to determine edge existence:

$$|W_{i,j}| \geq \epsilon_0 \iff (x_i, x_j) \in E(G(W)). \quad (14)$$

Similarly, path existence requires a threshold $\epsilon > 0$, leading to the relaxed condition $(p(W))_{i,j} \geq \epsilon$. This can be equivalently reformulated using an auxiliary function, as employed in the work by Wang et al. (2024):

$$(\bar{p}(W))_{i,j} = 0, \text{ where } \bar{p}(W) \equiv \text{ReLU} (\epsilon - p(W)). \quad (15)$$

Here, $\text{ReLU}(\cdot)$ is the element-wise ReLU function, defined as $\text{ReLU}(x) = \max(0, x)$. The equality $(\bar{p}(W))_{i,j} = 0$ is equivalent to enforcing $(p(W))_{i,j} \geq \epsilon$.

**Remark 2.** We refer to $\epsilon_0$ as the *edge threshold* and $\epsilon$ as the *path threshold* in the following discussion. Next, we analyze why the path threshold-based characterization $(\bar{p}(W))_{i,j} = 0$ fails to equivalently represent path existence.

**Issue of Non-Equivalence to Path Existence.** We show that Equation (15) is not equivalent to the path existence condition $x_i \rightsquigarrow x_j \in G(W)$ when $G(W)$ is constructed using edge relaxation in Equation (14).

To begin with, we assume that elements in $|W|$ are bounded due to regularization and data approximation:

**Assumption 1.** There exists a constant $\sigma > \epsilon_0$, where $\epsilon_0$ is the edge threshold, such that $|W|_{i,j} \leq \sigma$ for all $i, j$.

Under this assumption, we establish the following result:

**Lemma 1.** *(Sufficient Condition) There exists a finite threshold* $f(\epsilon_0, \sigma) > \epsilon_0$ *such that* $(\bar{p}(W))_{i,j} = 0$ *is sufficient to guarantee path existence* $x_i \rightsquigarrow x_j \in G(W)$ *under edge relaxation in Equation* (14) *if and only if* $\epsilon \geq f(\epsilon_0, \sigma)$.

This result indicates that if path threshold $\epsilon$ is not sufficiently large, the condition $(\bar{p}(W))_{i,j} = 0$ is insufficient to ensure the existence of path $x_i \rightsquigarrow x_j$. Additionally, we have:

**Lemma 2.** *(Necessary Condition) The continuous equality* $(\bar{p}(W))_{i,j} = 0$ *is necessary for the path existence* $x_i \rightsquigarrow x_j \in G(W)$ *under edge relaxation in Equation* (14) *if and only if* $\epsilon \leq \min(\epsilon_0, \epsilon_0^d)$.

This result implies that when path threshold $\epsilon$ is slightly larger, $(\bar{p}(W))_{i,j} = 0$ is no longer necessary for path existence, potentially introducing extraneous constraints.

Lemma 1 and Lemma 2 establish the critical path threshold $\epsilon$ values for the sufficiency and necessity of the continuous relaxation $\bar{p}(W) = 0$ in representing path existence. However, no single $\epsilon$ satisfies both conditions simultaneously, as $f(\epsilon_0, \sigma) > \min(\epsilon_0, \epsilon_0^d)$. Consequently, $\bar{p}(W) = 0$ in Equation (15) fails to equivalently represent path existence, underscoring the need for further refinement.

### 3.2. Binary-Masked Continuous Relaxation for Equivalent Path Existence Characterization

To equivalently represent the path existence constraint, we introduce the following function:

$$\hat{p}(W) \equiv \bar{p}(W) \circ b(W), \text{ where } \bar{p}(W) \text{ is defined in (15),}$$

$$\text{and } b(W) \equiv \mathbb{I}\left(\sum_{k=1}^{d} (\mathbb{I}(|W| \geq \epsilon_0))^k = 0\right). \quad (16)$$

Here, $\mathbb{I}(\cdot)$ is an element-wise indicator function that returns 1 if the inner condition holds and 0 otherwise, and $\circ$ denotes the element-wise product.

To interpret the overall operation of the function $\hat{p}(W)$, it selectively activates or deactivates path existence constraints in $\bar{p}(W)$ based on whether the constraint is already satisfied, as determined by the binary mask $b(W)$:

**Proposition 3.** *For* $b(W)$ *defined in Equation* (16) *and* $G(W)$ *constructed with edge threshold* $\epsilon_0$ *in Equation* (14),

$(b(W))_{i,j} = 0$ *if there exists at least one directed path from* $x_i$ *to* $x_j$ *in* $G(W)$, *and* $(b(W))_{i,j} = 1$ *if no such path exists.*

This result ensures that $\hat{p}(W)$ activates the influence of $(\bar{p}(W))_{i,j}$ only if no directed path exists in $G(W)$. This mechanism eliminates redundant constraints when path existence is already satisfied, thus preserving the necessity:

**Lemma 3.** *(Necessity) For any* $\epsilon$, *if there exists at least one directed path from* $x_i$ *to* $x_j$ *in* $G(W)$ *for* $G(W)$ *constructed by Equation* (14), *then* $(\hat{p}(W))_{i,j} = 0$.

Next, we establish the sufficiency condition when the path threshold $\epsilon$ is large enough, i.e., $\epsilon > f(\epsilon_0, \sigma)$. Consider the condition $(\hat{p}(W))_{i,j} = 0$:

$$(\hat{p}(W))_{i,j} = 0 \implies (b(W))_{i,j} \cdot (\bar{p}(W))_{i,j} = 0. \quad (17)$$

If $(b(W))_{i,j} = 0$, Proposition 3 ensures that $x_i \rightsquigarrow x_j \in G(W)$. If $(\bar{p}(W))_{i,j} = 0$, Lemma 1 guarantees that $x_i \rightsquigarrow x_j \in G(W)$. Thus, we obtain:

**Lemma 4.** *(Sufficiency) For some finite* $f(\epsilon_0, \sigma) > \epsilon_0$, *if* $\epsilon \geq f(\epsilon_0, \sigma)$, *then* $(\hat{p}(W))_{i,j} = 0$ *implies the existence of at least one directed path from* $x_i$ *to* $x_j$ *in* $G(W)$ *constructed with edge threshold* $\epsilon_0$ *in Equation* (14).

Lemma 3 and Lemma 4 together establish the equivalence of $(\hat{p}(W))_{i,j} = 0$ in characterizing path existence when $\epsilon$ is sufficiently large, as stated in the following result:

**Theorem 1.** *There exists at least one directed path from* $x_i$ *to* $x_j$ *in* $G(W)$ *constructed by Equation* (14) *if and only if* $(\hat{p}(W))_{i,j} = 0$, *where* $\hat{p}(W)$ *is defined by Equation* (16) *with* $\epsilon \geq f(\epsilon_0, \sigma)$ *for some finite* $f(\epsilon_0, \sigma)$.

In summary, we introduce a binary mask $b(W)$ that computes the path sum matrix using a binary adjacency matrix representing thresholded edge presence. This mask *accurately determines* whether a directed path already exists between two nodes, thereby selectively avoiding the redundant influence of the equality constraint $(\bar{p}(W))_{i,j} = 0$.

Essentially, this operation removes the constraint on $\epsilon$ for ensuring the necessity of path existence, making it valid for any $\epsilon$. Thus, a sufficiently large $\epsilon$ can be chosen to satisfy both necessity and sufficiency, achieving equivalence.

**Remark 3.** See Section B in the Appendix for details on the motivation and alternative explorations.

**Task Formulation.** Suppose the set of path existence constraints is represented by $A \in \{0,1\}^{d \times d}$, where $A_{i,j} = 1$ indicates the constraint $x_i \rightsquigarrow x_j \in G(W)$. The task of differentiable structure learning with $A$ is formulated as:

$$\min_{W} \|DW - D\|_2^{\ell} + \lambda \|W\|_1 + \gamma \sum (\hat{p}(W) \circ A),$$
$$\text{subject to } \hat{h}(W) = 0, \quad (18)$$

where $\gamma > 0$ is the weight of the path existence loss, and $\sum B$ represents $\sum_{i,j} B_{i,j}$.

In Equation (18), path existence constraints $A$ are treated as soft constraints by incorporating $(\hat{p}(W))_{i,j}$ as a loss term rather than enforcing $(\hat{p}(W))_{i,j} = 0$ strictly. This relaxation is necessary due to the inherent conflict between the path existence loss $(\hat{p}(W))_{i,j}$ and the acyclicity loss $\hat{h}(W)$ during optimization, as stated in the following result:

**Proposition 4.** *The losses $(\hat{p}(W))_{i,j}$ and $\hat{h}(W)$ exhibit a gradient conflict, consistently pushing the parameters of $W$ in opposite directions during optimization:*

$$\forall u, v, \quad \nabla_{W_{u,v}}\hat{h}(W) \cdot \nabla_{W_{u,v}}(\hat{p}(W))_{i,j} \leq 0. \quad (19)$$

When optimizing conflicting losses, the process may converge to a suboptimal compromise where neither loss is fully minimized. To prevent violations of the acyclicity constraint, the path existence term is incorporated as a loss rather than an equality constraint. This approach ensures $\hat{h}(W)$ is fully optimized, prioritizing the fundamental DAG assumption in structure learning over prior knowledge constraints.

# 4. Order-Guided Optimization with Path Existence Constraints

This section first analyzes the order-violation issue in optimizing the path existence-based structure learning problem in Equation (18). We then introduce an order-guided optimization strategy to address this challenge.

## 4.1. Issue of Path Order Violation

When solving the path existence-based structure learning problem in Equation (18), full adherence to path existence constraints may fail, leading to suboptimal solutions. This issue arises from two interrelated factors: 1) The path existence loss is highly sensitive to initial guesses, potentially introducing order-violating paths. 2) In the augmented Lagrangian process, the weight of the acyclicity loss increases over iterations, preventing the recovery of paths that violate the graph's existing order.

**Partial Order Relations.** The (partial) order of variables refers to their position in a DAG's topological ordering:

**Definition 2** (Topological Order). A topological order of a DAG is a permutation $\pi$ of nodes such that for every directed edge $(i, j)$ in the DAG, $\pi^{-1}(i) < \pi^{-1}(j)$, where $\pi^{-1}(x)$ denotes the position of $x$ in $\pi$.

**Definition 3** (Partial Order). The partial order relation between nodes $i$ and $j$, denoted as $i \prec j$, implies that $i$ precedes $j$ in the topological order, i.e., $\pi^{-1}(i) < \pi^{-1}(j)$.

By these definitions, the existence of a path $x_i \rightsquigarrow x_j \in G$

in a DAG implies a partial order $x_i \prec x_j$. The path order violation is subsequently defined as:

**Definition 4** (Path Order Violation). Two directed paths are in order violation if one path contains a sub-path that implies the reverse partial order of the other. In this case, the paths are said to be order-violating paths, or one path is said to violate the order of the other.

A recovered path $x_i \rightsquigarrow x_j$ from optimizing $(\hat{p}(W))_{i,j}$ may be in order violation with other paths specified in the constraint. For illustration, consider the gradient of $(\hat{p}(W))_{i,j}$ with respect to an arbitrary edge weight $W_{u,v}$:

$$\nabla_{|W_{u,v}|}(\hat{p}(W))_{i,j} = -\sum_{p \in \mathcal{P}_{i \rightsquigarrow j}^{(u,v)}} \prod_{e \in p \setminus \{(u,v)\}} |W_e|, \quad (20)$$

where $\mathcal{P}_{i \rightsquigarrow j}^{(u,v)}$ denotes the set of directed paths of length at most $k$ from $x_i$ to $x_j$ that include the edge $(x_u, x_v)$, and $W_e$ represents the weight of edge $e$ in $W$.

Equation (20) illustrates how the path existence loss $(\hat{p}(W))_{i,j}$ influences the optimization of an arbitrary edge $(x_u, x_v)$. If a nearly complete path $x_i \rightsquigarrow x_j$ exists except for the missing edge $(x_u, x_v)$, the loss strongly promotes the inclusion of $(x_u, x_v)$ in the structure. Consequently, the optimization of $(\hat{p}(W))_{i,j}$ becomes highly sensitive to initial conditions, as demonstrated in the following example:

**Example 1.** Consider two different initial weight matrices:

$$W_1 = \begin{bmatrix} 0 & 2 & 0.1 \\ 0.1 & 0 & 0.1 \\ 0.1 & 0.1 & 0 \end{bmatrix}, W_2 = \begin{bmatrix} 0 & 0.1 & 2 \\ 0.1 & 0 & 0.1 \\ 0.1 & 0.1 & 0 \end{bmatrix}. \quad (21)$$

With settings $\epsilon_0 = 1$ and $\epsilon = 9$, optimizing:

$$\min_W (\hat{p}(W))_{1,3} + 0.1\|W\|_1 \quad \text{s.t. } \forall i, W_{i,i} = 0,$$

using gradient descent $W^{t+1} = W^t - \alpha\nabla_W\mathcal{L}(W)$ with initial conditions $W_1$ and $W_2$ leads to:

$$W_1^{\text{opt}} = \begin{bmatrix} 0 & 1.0 & 0.0 \\ 0.0 & 0 & 1.0 \\ 0.0 & 0.0 & 0 \end{bmatrix}, W_2^{\text{opt}} = \begin{bmatrix} 0 & 0.0 & 1.0 \\ 0.0 & 0 & 0.0 \\ 0.0 & 0.0 & 0 \end{bmatrix}. \quad (22)$$

Here, different initial guesses lead to different recovered edges in the optimal solution of the path existence loss. In structure learning, this effect can override data approximation, unintentionally introducing edges (or paths) that violate the order of other paths. In the above example, if the path existence constraints include $x_1 \rightsquigarrow x_3$ and $x_3 \rightsquigarrow x_2$, then $W_1^{\text{opt}}$, where $x_1 \rightsquigarrow x_3$ contains the edge $(x_2, x_3)$, introduces an order $x_2 \prec x_3$, violating the path $x_3 \rightsquigarrow x_2$.

In the augmented Lagrangian method commonly used for structure learning, the acyclicity function $\hat{h}(W)$ is incorporated as a loss term with an increasing weight over iterations

to ensure full optimization to zero. Consequently, when the acyclicity weight becomes large, it prevents the recovery of paths that contain order-violating paths in the current structure, as stated below:

**Proposition 5.** *Suppose the current graph $G(W)$ contains a directed path from $x_j$ to $x_i$, but there is no path from $x_i$ to $x_j$. In this case, the loss function:*

$$\mathcal{L}(D, W) = F(W; D) + \lambda\|W\| + \gamma(\hat{p}(W))_{i,j} + \rho\hat{h}(W)$$

*prevents the addition of any edge $(u, v)$ that would establish a path from $x_i$ to $x_j$ in an updated graph $G'$, where $G' = (X, E(G) \cup \{(u, v)\})$, as long as $\rho$ is sufficiently large.*

This result shows that if $x_i \rightsquigarrow x_j$ conflicts with a reversed path in the current graph, it will be forbidden, even if it was originally part of path existence constraints.

For a broader perspective on the order-violation issue, the acyclicity loss weight is initially small, allowing order-violating paths to emerge due to the influence of initial guesses. As the acyclicity loss strengthens, one of the conflicting order-violating paths is eliminated, preventing the full satisfaction of the path existence constraint.

---

**Algorithm 1** Differentiable Structure Learning with Path Existence Constraints

---

**Require:** Data $D$, binary mask $A \in \{0,1\}^{d\times d}$ of path-existence constraints, edge threshold $\epsilon_0$

1: **Define** backbone model: $M\langle L, h, W_\theta\rangle$, with data-fit loss $L$, acyclicity loss $h$, and structure parameters $W_\theta$.

2: **Define** path existence loss:

$$L' = L + \sum(\hat{p}(W) \circ A)$$

3: **Define** order-based acyclicity loss:

$$h_o = h + \sum(p(W) \circ (A^+)^T)$$

4: Solve order-based optimization (initializing from zero):

$$W_o \leftarrow M_o(D, 0), \quad \text{where} \quad M_o\langle L, h_o, W_\theta\rangle$$

5: Solve path existence-based optimization using the order-based optimization result $W_o$ as initialization:

$$W_p \leftarrow M_p(D, W_o), \quad \text{where} \quad M_p\langle L', h, W_\theta\rangle$$

6: Threshold learned structure:

$$\bar{W}_p \leftarrow \mathbb{I}(|W_p| > \epsilon_0)$$

7: **Return** Final learned structure $\bar{W}_p$

---

### 4.2. Order-Guided Optimization

To address the order-violation issue in optimizing Equation (18), we propose an order-guided optimization strategy. Specifically, we aim to derive a weight matrix $W$ that satisfies all partial orders implied by the specified path existence constraints and then use this $W$ as the initial guess for the path existence-constrained problem in Equation (18).

Since each path existence constraint $x_i \rightsquigarrow x_j$ implies a partial order $x_i \prec x_j$, the set of partial orders, denoted as $O$, coincides with the path existence set. To enforce these constraints, we should ensure that each partial order $x_i \prec x_j$ is respected by forbidding its reversed path $x_j \rightsquigarrow x_i$. However, because partial orders are transitive while path absence is not, we must derive the complete set of partial orders using transitive closure:

**Definition 5** (Transitive Closure). The transitive closure of a set $S = \{(x, y)\}$, denoted by $S^+$, contains a pair $(a, b)$ if and only if there exists a sequence $a = x_0, x_1, \ldots, x_k = b$ such that $(x_i, x_{i+1}) \in S$ for all $i$.

With this, we can equivalently enforce the partial order set $O$ using a set of path absence constraints:

**Proposition 6** ((Ban et al., 2024, Proposition 2)). *Given a partial order set $O = \{x_i \prec x_j\}$, a DAG $G$ satisfies $O$ if and only if, for all $x_i \prec x_j$ in $O^+$, there exists no directed path from $x_j$ to $x_i$ in $G$.*

By combining this statement with the continuous characterization $(p(W))_{i,j} = 0$, defined in Equation (12) for path absence, differentiable structure learning with the partial order constraint set $O$ can be formulated as:

$$\min_{W \in \mathbb{R}^{d \times d}} \|DW - D\|_2^\ell + \lambda\|W\|_1,$$
$$\text{subject to } \hat{h}(W) = 0, \; p(W) \circ (A^+)^T = 0, \tag{23}$$

where $(\cdot)^T$ denotes matrix transposition, $A$ is the mask of the path existence set corresponding to $O$, and $p(W) \circ (A^+)^T = 0$ ensures the absence of all reversed paths associated with the indicated orders in the transitive closure $O^+$. The mask $A^+$, corresponding to $O^+$, is computed as:

$$A^+ = \mathbb{I}\left(\sum_{k=1}^d A^k > 0\right). \tag{24}$$

Solving Equation (23) yields an outcome that satisfies the partial order set $O$ implied by the path existence constraints. This outcome is then used as the initial guess for optimizing path existence-constrained structure learning problem (18).

An overall illustration of differentiable structure learning with path existence constraints is presented in Algorithm 1.

**Rationale Justification.** During optimization, the partial order loss and acyclicity loss are gradient-consistent, meaning that they always push parameters in $W$ in the same direction. This consistency enables a fuller optimization of both losses, ensuring that all partial orders implied by path existence constraints are satisfied.

With an initial guess that adheres to all partial orders of paths, the risk of generating order-violating paths is reduced. This is due to the acyclicity loss discouraging the recovery of paths with reversed orders relative to the initial guess. Consequently, the optimization of the path existence-constrained problem is guided toward recovering paths without violating the order of each other, leading to a stronger adherence to path existence conditions.

Furthermore, optimizing structure learning with path existence constraints while respecting all their partial orders ensures that no erroneous orders are introduced. This helps refine the optima space structure learning where each optima corresponds to a topological order:

**Proposition 7.** *Each optimal solution of the equality-constrained problem (5) corresponds to the global optimum of the following convex optimization problem:*

$$\min_{W \in \mathbb{R}^{d \times d}} \|D(W \circ M) - D\|_2^\ell + \lambda \|W\|_1, \quad (25)$$

*where $M_{i,j} = 1$ if $\pi(i) < \pi(j)$ and $M_{i,j} = 0$ otherwise, for a specific permutation $\pi$ of $\{1, 2, \ldots, d\}$.*

Hence, the optima space is refined to a more accurate one, thus improving the quality of the recovered structure.

In summary, the partial order-constrained initial guess enforces all partial orders implied by path existence constraints, thereby reducing the occurrence of order-violating paths through the acyclicity loss. The resulting solution, which adheres to all path existence constraints, avoids introducing erroneous orders, ultimately leading to a more accurate reconstruction of the true structure.

## 5. Experiments

We present partial experimental results and analysis here, with the full results available in the appendix.

**Synthetic Data.** Random DAGs are generated using Erdős-Rényi (ER) and scale-free (SF) models with node degrees in $\{2, 4\}$ and numbers of nodes $d$ in $\{10, 20, 30, 50\}$. For observational data, uniformly random weights are assigned to the weighted adjacency matrix $B$. Given $B$, samples are generated using the structural equation model $X = B^T X + z, X \in \mathbb{R}^d$, with noise models $\{$Gaussian (gauss), Exponential (exp)$\}$. The sample size is $n = \{20, 1000\}$. To generate constraints, we compute the path mask $W_P$ from the ground truth adjacency mask $W$ as

$W_P = \mathbb{I}(\sum_{k=1}^d W^k > 0)$, representing all paths. We then randomly select $q\%$ of these paths as constraints.

**Real-World Data.** We use real-world protein and phospholipid expression data from (Sachs et al., 2005), which measures interactions in human immune system cells. This dataset is a standard benchmark in graphical modeling due to its consensus network of 11 nodes and 17 edges, validated through experimental annotations.

**Methods.** The backbone algorithms include NOTEARS (Zheng et al., 2018; 2020), DAGMA (Bello et al., 2022), and GOLEM (Ng et al., 2022). Path existence (PE)-based structure learning is denoted as 'PE-*alg*', where *alg* represents the backbone algorithm used.

**Metrics.** We evaluate performance using structural Hamming distance (SHD), true positive rate (TPR), false discovery rate (FDR), F1 score, and path recovery rate (satisfied constraints / total constraints).

**Setup.** Default parameters: edge threshold $\epsilon_0 = 0.3$, path threshold $\epsilon = 10$, $L_1$ weight $\lambda = 0.1$, path existence weight $\gamma = 1$, path percentage $q = 80$. Other parameters follow defaults of backbone algorithms. Experiments run on an AMD Ryzen 9 7950X (4.5 GHz) CPU, NVIDIA RTX 3090 GPU, and 32 GB RAM.

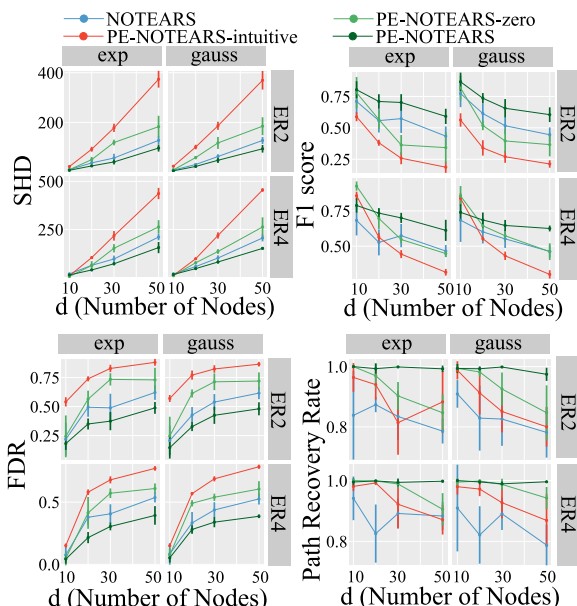

*Figure 1.* Comparison and ablation results of main metrics.

### 5.1. Comparison and Ablation Results

We compare our method (PE-NOTEARS) with standard NOTEARS (without path existence constraints) and two ab-

*Table 1.* Results of our method on the real-world Sach dataset with various sample sizes and PE constraint numbers.

| Method | Sachs-100 | | | | Sachs-200 | | | | Sachs-500 | | | | Sachs-902 | | | |
|---|---|---|---|---|---|---|---|---|---|---|---|---|---|---|---|---|
| | SHD | FDR | TPR | F1 | SHD | FDR | TPR | F1 | SHD | FDR | TPR | F1 | SHD | FDR | TPR | F1 |
| NOTEARS | 22 | 0.72 | 0.47 | 0.34 | 17 | 0.63 | 0.47 | 0.41 | 13 | 0.56 | 0.41 | 0.42 | 13 | 0.58 | 0.41 | 0.41 |
| PE-NOTEARS-25 | 20 | 0.67 | 0.52 | 0.40 | 15 | 0.60 | 0.47 | 0.43 | 13 | 0.55 | 0.47 | 0.45 | 12 | 0.52 | 0.47 | 0.47 |
| PE-NOTEARS-50 | 16 | 0.61 | 0.59 | 0.46 | 13 | 0.55 | 0.53 | 0.48 | 13 | 0.55 | 0.47 | 0.46 | 12 | 0.55 | 0.47 | 0.46 |
| PE-NOTEARS-75 | 16 | 0.61 | 0.59 | 0.46 | **13** | **0.55** | **0.53** | **0.48** | 12 | 0.55 | 0.53 | 0.48 | **11** | **0.52** | **0.53** | **0.50** |
| PE-NOTEARS-100 | **15** | **0.60** | **0.59** | **0.47** | 14 | 0.57 | 0.53 | 0.47 | **11** | **0.50** | **0.58** | **0.54** | 12 | 0.55 | 0.53 | 0.48 |

lation variants: PE-NOTEARS-zero, which uses the binary-masked loss $\hat{p}$ but initializes with a zero matrix, and PE-NOTEARS-intuitive, which directly applies the continuous relaxed loss $\bar{p}$ from Equation (15) with a zero initial guess.

Results in Figure 1 show that our method significantly outperforms the baseline and ablation variants. It achieves lower SHD and higher F1 score, indicating better overall quality. The FDR is lower, indicating less erroneous edge recovery, while the path recovery rate is higher, ensuring stronger adherence to path existence constraints. A lower FDR confirms that using $\hat{p}$ mitigates the redundancy issue in $\bar{p}$ for path existence characterization. A higher path recovery rate highlights the effectiveness of the order-guided initial guess in resolving order-violation issues in recovered paths.

### 5.2. Results on Real-World Data

We evaluate our method on the real-world Sachs dataset, applying different percentages $p$ of path existence constraints, denoted as PE-NOTEARS-$p$. Using a subset of Sachs's cell data with 902 samples, we vary the sample size $n$, denoted as Sachs-$n$, by selecting the first $n$ samples.

Table 1 shows that our method consistently outperforms the data-driven baseline, with greater improvements as more constraints are incorporated. Notably, using only 100 samples, our method achieves a higher F1 score than the baseline trained on all 902 samples, demonstrating its potential for reducing experimental resource requirements.

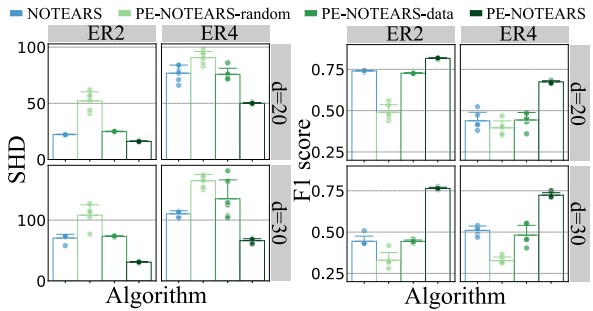

*Figure 2.* Ablation results on various initialization strategies.

### 5.3. Ablations on Order-Guided Optimization

We conduct an ablation study on different initialization strategies to assess the effectiveness and robustness of order-guided optimization. The evaluated strategies include: PE-NOTEARS using the partial order-guided outcome, PE-NOTEARS-random with a random initial guess, and PE-NOTEARS-data using the NOTEARS outcome (without prior) as the initial guess.

For each method, we randomly generate six initial guesses and report their individual results on the same dataset, with partial results shown in Figure 2. The partial order-guided initialization achieves the best performance (lowest SHD, highest F1) with the highest stability (lowest variance), demonstrating its robustness and effectiveness.

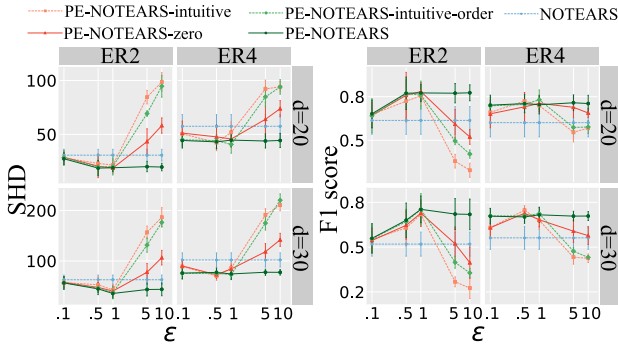

*Figure 3.* Ablation results with varying path thresholds.

### 5.4. Ablations with Varying $\epsilon$ Settings

We conduct an ablation study on varying path thresholds $\epsilon$, a key hyperparameter for the path existence loss. The evaluated methods include PE-NOTEARS, PE-NOTEARS-zero, and PE-NOTEARS-intuitive, as introduced earlier, along with a new variant, PE-NOTEARS-intuitive-order, which applies $\bar{p}$ as the PE loss with the order-guided initial guess for a more comprehensive ablation. Results for $\epsilon$ values ranging from 0.1 to 10 are shown in Figure 3.

Our method consistently outperforms the ablation versions across different $\epsilon$ settings. Performance initially improves

as $\epsilon$ increases, ensuring the sufficiency of $\hat{p}$ in representing path existence, and then stabilizes once $\hat{p}$ fully aligns with path existence. In contrast, other ablation variants show initial improvement but degrade as $\epsilon$ increases, particularly for $\bar{p}$ (dotted lines), which loses necessity at large $\epsilon$ (Lemma 2), resulting in excessive extra edges. Additionally, versions without order-guided optimization (red-series lines) exhibit greater variance due to the sensitivity of the path existence loss to initial guesses. These results align with our analysis.

### 5.5. Other Results and Analysis

The appendix provides additional evaluations on time complexity, path absence constraints, nonlinear data, backbone variations, and the effect of varying constraint numbers. Furthermore, supplementary results covering additional metrics, different graph settings, and ablation studies related to the experiments in this section are also included.

## 6. Discussion

### 6.1. Limitations and Future Directions

The current path existence formulation lacks explicit mechanisms to prevent path order violation and is computationally slower than acyclicity losses. Future work may explore more efficient and reliable characterization approaches.

### 6.2. Conclusion

This paper extends differentiable structure learning by integrating ancestral constraints, enabling the use of abstract prior knowledge to enhance fine-grained causal discovery. We identify key challenges in incorporating path existence constraints into continuous optimization and propose a binary-masked characterization with an order-guided optimization strategy to address them. Theoretical analysis supports the correctness of our approach, which is validated on both synthetic and real-world datasets.

## Impact Statement

The proposed ancestral constraint-based differentiable structure learning framework combines the expressive power of neural networks and GPU-accelerated optimization with the reliability of knowledge-guided constraints. This integration enables efficient and accurate recovery of authentic, previously unknown causal mechanisms.

The algorithm has broad applicability in scientific discovery domains, where high-level prior knowledge about variable interactions can guide the discovery of fine-grained causal structures. Such domains include biology, medicine, and social sciences, where partial knowledge is common but direct causal relationships remain elusive.

Beyond scientific applications, this work contributes to the broader AI landscape by advancing NOTEARS-based causal analysis frameworks. Its ideas may benefit a wide range of downstream tasks in fields such as computer vision, fault diagnosis, and multi-agent systems, where causal reasoning plays an increasingly central role.

## Acknowledgements

This research was supported in part by the National Key R&D Program of China (No. 2021ZD0111700), in part by the National Nature Science Foundation of China (No. 62406302, 62137002, 62176245), in part by the Natural Science Foundation of Anhui province (No. 2408085QF195), in part by the Fundamental Research Funds for the Central Universities under Grant WK2150110035.

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

# Appendix

# A. Related Work

This section reviews relevant studies related to this paper. We first provide a brief overview of traditional structure learning methods and the integration of prior structural constraints. We then discuss differentiable structure learning approaches and studies incorporating prior constraints in this field.

## A.1. Traditional Structure Learning and Prior Constraint Integration

### A.1.1. CAUSAL GRAPHICAL MODELS

Causal models are categorized based on graph type, including DAGs, undirected graphs, and mixed graphs.

Causal DAGs primarily encompass Bayesian networks (BNs) (Pearl, 2000) and structural equation models (SEMs) (Spirtes et al., 2001). BNs represent causal relationships through conditional probabilities, $P(x_i \mid \mathrm{Pa}_i^G)$, while SEMs model causality via continuous functions, $x_i = f(\mathrm{Pa}_i^G, z_i)$, where $z_i$ is a noise term. This paper conducts experiments with additive noise models (ANMs), a subclass of SEMs where the noise term is additive, i.e., $x_i = f(\mathrm{Pa}_i^G) + z_i$.

Undirected causal graphs primarily involve Markov Random Fields, which capture symmetric dependencies without causal direction and are commonly used for modeling spatial or network-based relationships (Hinton et al., 2005).

Mixed graphs allow the coexistence of directed, undirected, and bidirected edges. These include maximal ancestral graphs (MAGs) (Richardson & Spirtes, 2002), used to represent causal relationships in the presence of latent variables or selection bias; ancestral graphs (Zhang, 2008), which generalize both DAGs and MAGs; and partial DAGs (Andersson et al., 1997), which use undirected edges to represent ambiguous causal relationships.

### A.1.2. TRADITIONAL STRUCTURE LEARNING.

Structure learning of causal DAGs aims to identify the model structure that best represents the distributions of observational data (Scutari et al., 2019). Given the DAG constraint, traditional methods typically employ discrete approaches, including constraint-based, score-based, and hybrid methods.

Constraint-based methods reconstruct the graph using conditional independence tests, followed by edge orientation within the DAG constraint. Key algorithms include the Peter and Clark (PC) algorithm (Spirtes & Glymour, 1991) and the Fast Causal Inference (FCI) algorithm (Spirtes et al., 1995).

Score-based methods evaluate DAGs using scoring functions that measure goodness-of-fit to data, followed by a search for an optimal DAG. Popular scoring functions include Bayesian Information Criterion (BIC) (Schwarz, 1978), Bayesian Dirichlet Equivalent Uniform (BDeu) (Heckerman & Geiger, 1995), and Minimum Description Length (MDL) (Rissanen, 1978). Search algorithms include Greedy Equivalence Search (GES) (Chickering, 2002), Hill Climbing (HC) (Neapolitan et al., 2004), Simulated Annealing (Friedman et al., 1997), Tabu search (Li & Beek, 2018), etc.

Hybrid methods combine constraint-based and score-based techniques. Notable examples include Max-Min Hill-Climbing (MMHC) (Tsamardinos et al., 2006) and Partitioned Hybrid Greedy Search (PHGS) (Huang & Zhou, 2022).

### A.1.3. PRIOR CONSTRAINT INTEGRATION.

Structural constraints representing prior causal knowledge primarily fall into three categories:

1. **Edge constraints** specify the presence or absence of edges in the graph. While straightforward to integrate into structure learning, they are often difficult to obtain, as they directly reveal causal interactions, the very objective of structure learning. Relevant studies include (Amirkhani et al., 2016; de Campos & Castellano, 2007).

2. **Ancestral constraints** specify the existence of certain directed paths or the absence of all directed paths between variables. These constraints capture abstract causal relationships that may be realized either directly or indirectly. Ancestral constraints provide a robust way to encode prior causality, aligning well with the abstract nature of knowledge. Related studies include (Li & Beek, 2018; Chen et al., 2016).

3. **(Partial) order constraints** define the ordering relationships among a subset (or all) of the variables. They are less

restrictive than edge and ancestral constraints, as they only enforce topological relationships without necessarily implying causal interactions. Given a total ordering, structure learning is reduced to a set of straightforward polynomial-time problems, so we primarily focus on the integration of partial order constraints. Partial order constraints are commonly integrated into order-based search algorithms, which assume a total variable ordering and then search for the optimal ordering that yields the globally optimal DAG (Teyssier & Koller, 2005; Heckerman et al., 1995). Partial order constraints help reduce the search space by eliminating conflicting orderings (Li & Beek, 2018; Deng et al., 2023a).

The integration of structural constraints into structure learning follows two main approaches: hard and soft integration. Hard integration strictly enforces constraints by rejecting graphs that violate them. The aforementioned studies predominantly follow this approach. Soft integration balances adherence to prior constraints with goodness-of-fit to data, typically by modifying the scoring function to penalize constraint violations. CaMML (O'Donnell et al., 2006) is an open-source causal discovery tool that employs soft constraint integration. A comprehensive review of prior constraint integration methods can be found in (Constantinou et al., 2023). Interestingly, both hard and soft integration methods have direct counterparts in the context of differentiable structure learning, as discussed in the main text.

### A.2. Differentiable Structure Learning and Prior Constraint Integration

#### A.2.1. DIFFERENTIABLE STRUCTURE LEARNING.

Zheng et al. first introduced differentiable structure learning, extending score-based structure learning task:

$$\max_G \sigma(G; D), \quad \text{subject to } G \in \text{DAG}. \tag{26}$$

For continuous causal graphical models such as SEMs, the score function $\sigma(G; D)$ can be formulated continuously using a weighted adjacency matrix $W \in \mathbb{R}^{d \times d}$ to represent the structure, leaving the DAG constraint as the only discrete component. To address this, Zheng et al. proposed a smooth acyclicity characterization using the equality constraint:

$$h(W) \equiv \text{Trace}(e^{W \circ W}) - d = 0. \tag{27}$$

This formulation transforms structure learning into a continuous equality-constrained optimization problem:

$$\min_{W \in \mathbb{R}^{d \times d}} \|F_\theta(W; D) - D\|_2^l + \lambda \|W\|, \quad \text{subject to } h(W) = 0. \tag{28}$$

The augmented Lagrangian method is used to solve this problem, gradually increasing the weight of the acyclicity loss $h(W)$ over iterations to ensure convergence to a DAG. Optimization stops when $h(W)$ falls below a predefined threshold ($10^{-8}$ by default) or when the loss weight reaches a maximum ($10^{16}$ by default). A threshold $\epsilon_0$ is then applied to determine the presence of edges in the learned structure. This optimization framework has been widely adopted in subsequent differentiable structure learning approaches.

Existing research on differentiable structure learning can be categorized into four main directions:

**1. Novel Learner Architectures $\theta$.** Zheng et al. introduced NOTEARS-MLP, an extension using multilayer perceptrons for nonlinear structure learning. Yu et al. proposed DAG-GNN, a generative model leveraging graph neural networks and data reconstruction properties. Other studies have explored reinforcement learning (Zhu et al., 2020) and adversarial neural networks (Kalainathan et al., 2022) for differentiable structure learning.

**2. Novel Acyclicity Characterizations $h(W)$.** Yu et al. proposed a polynomial-based acyclicity function as an alternative to the exponential formulation in NOTEARS. Wei et al. generalized polynomial-based characterizations using path (cycle) absence principles. Lee et al. showed that polynomial-based constraints are equivalent to constraining the spectral radius to zero and introduced a spectral radius-based acyclicity formulation. Bello et al. proposed DAGMA, a log-determinant-based acyclicity characterization leveraging the nilpotency property of DAGs.

**3. Novel Loss Formulations $\mathcal{L}(W, D)$.** Ng et al. introduced GOLEM, a likelihood-based loss function that replaces the least squares loss in NOTEARS. By incorporating soft sparsity and DAG penalties, GOLEM eliminates the need for a hard equality constraint, simplifying and accelerating DAG learning. Reisach et al. analyzed the effect of data variance on differentiable structure learning, introducing varsortability as a criterion to infer variable ordering. Their findings

suggest that performance improvements may stem from the increasing marginal variance of descendant nodes. Deng et al. further investigated this issue and proposed a likelihood-based scoring function with quasi-MCP regularization, ensuring scale-invariant properties.

**4. Applications in Causal Discovery.** Some studies extend NOTEARS to broader causal discovery settings. Brouillard et al. explored differentiable causal discovery in interventional and observational mixtures, a common real-world scenario. Bhattacharya et al. extended the framework to settings with unobserved confounders. Gao et al. addressed structure learning with missing data. Pamfil et al. applied differentiable causal discovery to dynamic time-series data. Some extensions apply structure learning in other machine learning fields, such as multi-task learning (Chen et al., 2021), federated learning (Ng & Zhang, 2022), representation learning (Yang et al., 2021), and transportable learning (Berrevoets et al., 2023).

Despite these advances, limited research has explored the integration of prior structural constraints in differentiable structure learning, particularly beyond simple edge constraints. The next section discusses this gap.

A.2.2. PRIOR CONSTRAINT INTEGRATION IN DIFFERENTIABLE STRUCTURE LEARNING.

For a long period, studies of differentiable structure learning with prior knowledge only focus on the edge constraints. Integrating the edge absence constraint can be implemented by directly freezing the corresponding parameters to 0 (a hard constraint), or adding a loss penalizing $|W|_{i,j}$ when forbidding $(x_i, x_j)$. For the edge existence, one can use the continuous relaxed equality $\text{ReLU}(\epsilon_0 - |W|_{i,j}) = 0$ to force the existence of edge $(x_i, x_j)$, or adding the loss of $\text{ReLU}(\epsilon_0 - |W|_{i,j})$. Here are some related studies (Hasan & Gani, 2022; Sun et al., 2023; Wang et al., 2024; Ma et al., 2024).

For more complex constraints over edge constraints, a recent work by Ban et al. integrates the partial order constraints in differentiable structure learning. The authors also formulate the partial order constraints as a set of path absence constraints, as shown in Equation (23). They further propose an efficient characterization typically handling long sequential orderings. We directly use the path absence-based characterization in this paper because the partial order constraints implied by path existences do not typically form long sequences. Some studies propose order-based optimization methods for structure learning, which can also integrate partial order constraints (Shahverdikondori et al., 2024; Deng et al., 2023b), despite in a different task formulation of NOTEARS. Wang et al. study incorporating various structural constraints into differentiable structure learning. However, the authors does not focus on the generalized forms of partial order or ancestral constraints, who assume the total ordering for the order constraint which is straightforward and assume the knowledge of path lengths for path existence constraints. Besides, their proposed characterization of path existence is still limited by the issue of Non-Equivalence to represent path existence, as illustrated in Section 3.1 of the main text.

In comparison, this paper addresses the generalized form of path existence constraints without knowing any information of the path. We propose an equivalent continuous characterization of the path existences with theoretical justification. We further analyze the path order violation issue typically optimizing the problem with path existence constraints, and introduce a stable optimization strategy by taking the solution of path existence-implied partial order-constrained structure learning as the initial guess. In a word, this work first systematically discusses and addresses differentiable structure learning with ancestral constraints, typically on the challenges related to continuous path existence constraints.

## B. Motivation of Binary-Masked Characterization

This section explains the motivation behind our binary-masked characterization for representing path existence constraints. Additionally, we discuss alternative approaches that were explored but ultimately failed, providing richer insight.

### B.1. Motivation

The intuitive characterization of path existence is given by $(\bar{p}(W))_{i,j} = 0$, where:

$$\bar{p}(W) \equiv \text{ReLU}\left(\epsilon - \sum_k^d |W|^k\right). \tag{29}$$

We have shown that this formulation cannot equivalently represent path existence for any choice of the path threshold $\epsilon$. If $\epsilon$ is too small, the edge weights along a path may not reach the edge threshold $\epsilon_0$, causing the path to be considered absent. Conversely, if $\epsilon$ is too large, the constraint equality cannot be satisfied even when the path existence constraint is satisfied.

Despite this theoretical limitation, practical experiments indicate that when $\epsilon$ is set to an appropriate intermediate value, the algorithm still performs reasonably well (see Figure 3 in the main text). However, the optimal choice of $\epsilon$ is unknown without access to the ground truth. The remaining aim is eliminating its dependence on an unpredictable and unstable $\epsilon$.

Motivated by this idea, we propose removing the redundant influence of $\bar{p}(W)$ once the path existence constraint is already satisfied. If this can be achieved, we can set $\epsilon$ to a sufficiently large value to equivalently enforce path existence. This approach is feasible because the exact conditions for edge existence are known, allowing us to derive a binary mask indicating path presence.

A natural question arises: if we can compute path presence as a matrix $b(W)$, why not directly use it to constrain path existence instead of relying on $\bar{p}(W)$? Unfortunately, this approach fails in an optimization setting. When a path is absent, the loss function $b(W)$ cannot propagate gradients to edge weights due to the ReLU function, which blocks gradient flow when its argument is negative.

In summary, using $b(W)$ alone cannot enforce path existence when a path is missing, which can only enforce path absence when a path is already present. This limitation explains why our approach integrates $\bar{p}(W)$ with the binary mask to achieve a robust path existence characterization.

### B.2. Alternative Explorations

We also explored a more concise path existence characterization by constructing the path existence loss in a dual manner to path absence. While theoretically appealing, this approach encountered practical challenges.

We begin by recalling the path absence loss:

$$(p(W))_{i,j} = \sum_{k=1}^{d} \sum_{p \in \mathcal{P}_{i,j}^k} \prod_{(q,s) \in p} |W|_{q,s} = 0, \tag{30}$$

where $\mathcal{P}_{i,j}^k$ represents the set of all paths from $x_i$ to $x_j$ with length $k$.

Interpreting this loss from a logical perspective, we consider $|W|_{i,j} = 0$ as an indicator of edge absence and $|W|_{i,j} > 0$ as an indicator of edge presence. Using numerical operations to represent logical expressions, where multiplication corresponds to logical **and** ($\wedge$) and addition corresponds to logical **or** ($\vee$), we can rewrite the loss as:

$$\vee_{k=1}^{d} \left( \vee_{p \in \mathcal{P}_{i,j}^k} \left( \wedge_{(q,s) \in p} (|W|_{q,s} > 0) \right) \right). \tag{31}$$

Here, the inner expression $\wedge_{(q,s) \in p}(|W|_{q,s} > 0)$ enforces that all edges in $p$ must exist, ensuring the presence of a path. The outer logical **or** operations ensure that if at least one path from $i$ to $j$ exists, the entire expression evaluates to **true**. Consequently, when this expression is **false** (numerically, $(p(W))_{i,j} = 0$), no path exists from $x_i$ to $x_j$.

To enforce path existence, we considered imposing the following reversed constraint:

$$\text{ReLU}\left( \epsilon - (p(W))_{i,j} \right) = 0. \tag{32}$$

However, this approach suffers from the *non-equivalence issue* discussed in the main text. Logically, this discrepancy arises because it does not accurately capture the existence condition for each edge, resulting in a mismatch between the numerical loss and the logical presence of a path.

To resolve this, we sought a more precise condition for edge existence:

$$\bar{E}_{i,j} = \text{ReLU}(\epsilon_0 - |W|_{i,j}), \tag{33}$$

where $\bar{E}_{i,j} = 0$ indicates the presence of edge $(i, j)$, and $\bar{E}_{i,j} > 0$ indicates its absence. Notably, we define this as a complementary matrix to allow gradient propagation to edge weights when an edge is missing, which is particularly useful for enforcing path existence.

Now, consider the following logical expression:

$$\wedge_{k=1}^{d} \left( \wedge_{p \in \mathcal{P}_{i,j}^k} \left( \vee_{(q,s) \in p} (\bar{E}_{q,s} > 0) \right) \right). \tag{34}$$

This expression states that for all paths $p$ from $x_i$ to $x_j$, at least one edge in $p$ is absent (i.e., $\bar{E}_{q,s} > 0$ for some $(q,s) \in p$). This is precisely the *path absence condition*. Therefore, its logical negation corresponds to the *path existence condition*.

By replacing logical operations with numerical ones, we derive the following continuous formulation:

$$(p^\star(W))_{i,j} = \prod_{k=1}^{d} \prod_{p \in \mathcal{P}_{i,j}^k} \sum_{(q,s) \in p} \text{ReLU}(\epsilon_0 - |W|_{q,s}). \tag{35}$$

This leads to the following theoretical result:

**Theorem 2.** *A directed path from $x_i$ to $x_j$ exists in $G(W)$ constructed using edge threshold $\epsilon_0$ if and only if $(p^\star(W))_{i,j} = 0$, where $p^\star$ is defined by Equation* (35).

Thus, we obtain a concise, continuous, and equivalent formulation for enforcing path existence constraints. However, despite its theoretical soundness, this formulation suffers from *numerical instability* in practice. The product-based loss is prone to *vanishing gradients*, making optimization difficult. Additionally, the magnitude of $(p^\star(W))_{i,j}$ varies unpredictably: for $\epsilon_0 > 1$, it may becomes excessively large, while for $\epsilon_0 < 1/d$, it collapses to near zero, losing its effectiveness in guiding optimization. These practical issues also motivated our final binary-masked characterization, which maintains equivalence while avoiding numerical instability.

## C. Proof of Statements

**Proposition 1.** *If there exists a directed path $(x_i, x_j)$ in a directed graph $G$ with $d$ nodes, then there exists at least one length-$k$ path $(x_i, x_j)$ in $G$ such that $k \leq d$.*

*Proof.* Suppose, for the sake of contradiction, that the *shortest* directed path from $x_i$ to $x_j$ has length $k$ with $k > d$. Label this path

$$p = (l_1, l_2, \ldots, l_{k+1}),$$

where $l_1 = x_i$ and $l_{k+1} = x_j$. Since the path has $k$ edges, it visits $k + 1$ vertices. Our assumption $k > d$ implies

$$k + 1 > d + 1,$$

so the path visits at least $d + 2$ (not necessarily distinct) vertices. But $G$ itself has only $d$ vertices in total. By the pigeonhole principle, at least one vertex must repeat in the *interior* of the path; formally, there exist indices

$$1 \leq s < t \leq k + 1$$

such that $l_s = l_t$ and $s \neq 1, t \neq k + 1$. (In other words, the repeated node is not just the start or the end.)

The subpath

$$(l_s, l_{s+1}, \ldots, l_t)$$

thus forms a directed cycle. We can "shortcut" the path $p$ by removing that cycle, obtaining a strictly shorter path from $x_i$ to $x_j$. This contradicts our choice of $p$ as the *shortest* directed path. Consequently, no shortest path can exceed length $d$. Therefore, if there exists a path at all, there must be one of length at most $d$. $\square$

**Corollary 1.** *(Theorem 1 in (Wei et al., 2020)) A directed graph $G(W)$ is a DAG if and only if $\hat{h}(W) = 0$ for any $\hat{h}$ defined in Equation* (11).

*Proof.* Equation (10) shows that $G$ is DAG equals that $\sum_{k=1}^{d} \text{Trace}\left(|W|^k\right) = 0$. Given the non-negativity of $\text{Trace}(|W|^k)$, we have that this condition also equals $\sum_{k=1}^{d} c_k \text{Trace}\left(|W|^k\right) = 0$ for $c_k > 0$. $\square$

**Proposition 2.** *For a graph $G(W)$ where edge existence follows $W_{i,j} \neq 0 \iff (x_i, x_j) \in E(G(W))$, at least one directed path from $x_i$ to $x_j$ exists if and only if $(p(W))_{i,j} > 0$, where $p(W)$ is defined in Equation* (12).

*Proof.* Equation (13) already indicates this result. $\square$

**Lemma 1.** *(Sufficient Condition) There exists a finite threshold $f(\epsilon_0, \sigma) > \epsilon_0$ such that $(\bar{p}(W))_{i,j} = 0$ is sufficient to guarantee path existence $x_i \rightsquigarrow x_j \in G(W)$ under edge relaxation in Equation* (14) *if and only if $\epsilon \geq f(\epsilon_0, \sigma)$.*

*Proof.* We aim to prove the statement

$$(\bar{p}(W))_{i,j} = 0 \implies (x_i \rightsquigarrow x_j) \in G(W) \quad \text{if and only if} \quad \epsilon \geq f(\epsilon_0, \sigma) > \epsilon_0, \tag{36}$$

where $f(\epsilon_0, \sigma)$ is some finite constant strictly exceeding $\epsilon_0$.

**Step 1. Construction of $f(\epsilon_0, \sigma)$.** Consider a situation in which $G(W)$ *does not* have a path from $x_i$ to $x_j$. Then by definition of

$$(p(W))_{i,j} = \Big(\sum_{k=1}^{d} |W|^k\Big)_{i,j},$$

there must exist at least one edge in *every* candidate path (of length $k \leq d$) with absolute weight $< \epsilon_0$. Consequently,

$$(p(W))_{i,j} = \Big(\sum_{k=1}^{d} |W|^k\Big)_{i,j} < \sum_{k=1}^{d} \Big|p_{i,j}^k\Big| \cdot \epsilon_0 \cdot \sigma^{k-1}, \tag{37}$$

where $p_{i,j}^k$ is an arbitrary $k$-length directed path from $x_i$ to $x_j$ that contains no self-loop edges $(x_m, x_m)$, and $\big|p_{i,j}^k\big|$ denotes the number of such $k$-length paths. Here, at least one edge on each $p_{i,j}^k$ must have weight $< \epsilon_0$, and we let $\sigma \leq 1$ (or some suitable parameter) capture potential reductions in absolute weights for the remaining edges.

The right-hand side of (37) is a finite quantity (depending on $d$, $\epsilon_0$, $\sigma$, and the number of $k$-paths). Hence, we can set

$$f(\epsilon_0, \sigma) = \sup \Big\{ (p(W))_{i,j} \;\Big|\; \neg(x_i \rightsquigarrow x_j) \in G(W)\Big\}.$$

Thus $f(\epsilon_0, \sigma)$ is *finite* and moreover

$$f(\epsilon_0, \sigma) \leq \sum_{k=1}^{d} \Big|p_{i,j}^k\Big| \epsilon_0 \, \sigma^{k-1}.$$

We must also show that $f(\epsilon_0, \sigma) > \epsilon_0$.

**Step 2. Verifying $f(\epsilon_0, \sigma) > \epsilon_0$.** We construct a specific $W^-$ such that $G(W^-)$ does *not* have a directed path from $x_i$ to $x_j$ but $(p(W^-))_{i,j} > \epsilon_0$. For instance, define

$$W_{i,j}^- = \epsilon_0 - \epsilon^-, \qquad W_{i,l}^- = \epsilon_0 - \epsilon^-, \qquad W_{l,j}^- = \sigma,$$

for some node $x_l$ $(l \neq j)$, where $0 < \epsilon^- < \epsilon_0 - \frac{\epsilon_0}{1+\sigma}$. Set all other entries $W_{u,v}^- = 0$. In this setup:

$$(p(W^-))_{i,j} = \Big(\sum_{k=1}^{d} |W^-|^k\Big)_{i,j} = W_{i,j}^- + W_{i,l}^- W_{l,j}^- = (\epsilon_0 - \epsilon^-)(1 + \sigma).$$

Since $(1 + \sigma)(\epsilon_0 - \epsilon^-) > \epsilon_0$ by choice of $\epsilon^-$, we get

$$(p(W^-))_{i,j} > \epsilon_0.$$

Yet $G(W^-)$ has no directed path from $x_i$ to $x_j$ (for instance, if $x_l$ does not lead to $x_j$ or if additional zeros cut off the path). Hence $(p(W^-))_{i,j} \in \{ (p(W))_{i,j} \mid \neg(x_i \rightsquigarrow x_j) \in G(W)\}$, so in particular

$$f(\epsilon_0, \sigma) \geq (p(W^-))_{i,j} > \epsilon_0.$$

**Step 3. Showing the Sufficient Condition.** By definition of $\bar{p}(W)$,

$$(\bar{p}(W))_{i,j} = \text{ReLU}\Big(\epsilon - (p(W))_{i,j}\Big).$$

If $(\bar{p}(W))_{i,j} = 0$, then
$$\epsilon - (p(W))_{i,j} \leq 0 \quad \implies \quad (p(W))_{i,j} \geq \epsilon.$$

Suppose now $\epsilon \geq f(\epsilon_0, \sigma)$. Then
$$(p(W))_{i,j} \geq \epsilon \geq f(\epsilon_0, \sigma),$$

implies that $(p(W))_{i,j}$ *cannot* lie in the set $\{(p(W))_{i,j} \mid \neg(x_i \rightsquigarrow x_j) \in G(W)\}$, because the supremum (over graphs with no $x_i x_j$ path) is $f(\epsilon_0, \sigma)$. Hence, $(p(W))_{i,j}$ must belong to the complementary set in which $x_i \rightsquigarrow x_j$ in $G(W)$. Therefore, when $\epsilon \geq f(\epsilon_0, \sigma)$,
$$(\bar{p}(W))_{i,j} = 0 \implies (p(W))_{i,j} \geq \epsilon \geq f(\epsilon_0, \sigma) \implies x_i \rightsquigarrow x_j \in G(W).$$

Thus $(\bar{p}(W))_{i,j} = 0$ indeed *implies* the path existence $x_i \rightsquigarrow x_j$, establishing the "sufficiency" part under the condition $\epsilon \geq f(\epsilon_0, \sigma)$.

**Step 4. Converse Argument.** If $\epsilon < f(\epsilon_0, \sigma)$, we can select a $W$ from the set $\{(p(W))_{i,j} \mid \neg(x_i \rightsquigarrow x_j) \in G(W)\}$ with $(p(W))_{i,j} > \epsilon$. For that $W$, we have
$$\epsilon < (p(W))_{i,j} \implies \epsilon - (p(W))_{i,j} < 0 \implies (\bar{p}(W))_{i,j} = 0,$$

yet by construction $G(W)$ has no directed path from $x_i$ to $x_j$. This contradicts the statement that $(\bar{p}(W))_{i,j} = 0$ would imply $x_i \rightsquigarrow x_j$. Hence $\epsilon$ must be at least $f(\epsilon_0, \sigma)$ for the "sufficiency" property to hold.

This completes the proof of Lemma 1. $\qquad\square$

**Lemma 2.** *(Necessary Condition) The continuous equality $(\bar{p}(W))_{i,j} = 0$ is necessary for the path existence $x_i \rightsquigarrow x_j \in G(W)$ under edge relaxation in Equation* (14) *if and only if $\epsilon \leq \min(\epsilon_0, \epsilon_0^d)$.*

*Proof.* Recall that $(|W|^k)_{i,j}$ can be interpreted as the *sum* of products of absolute weights over all possible directed paths of *length* $k$ from $x_i$ to $x_j$. Suppose every directed edge in $G(W)$ has absolute weight at least $\epsilon_0$. Consider a specific path from $x_i$ to $x_j$ of length $k$. The product of its edge-weights (in absolute value) is then $\geq \epsilon_0^k$. Summing over all path lengths $1 \leq k \leq d$, we get
$$\left(\sum_{k=1}^{d} |W|^k\right)_{i,j} \geq \min\left(\epsilon_0, \epsilon_0^2, \ldots, \epsilon_0^d\right) = \begin{cases} \epsilon_0^d, & \text{if } 0 < \epsilon_0 < 1, \\ \epsilon_0, & \text{if } \epsilon_0 \geq 1. \end{cases}$$

Hence, in either case,
$$\left(\sum_{k=1}^{d} |W|^k\right)_{i,j} \geq \min(\epsilon_0, \epsilon_0^d).$$

By definition,
$$(\bar{p}(W))_{i,j} = \text{ReLU}\left(\epsilon - \left(\sum_{k=1}^{d} |W|^k\right)_{i,j}\right).$$

If $\epsilon \leq \min(\epsilon_0, \epsilon_0^d)$, then
$$\epsilon - \left(\sum_{k=1}^{d} |W|^k\right)_{i,j} \leq \min(\epsilon_0, \epsilon_0^d) - \min(\epsilon_0, \epsilon_0^d) = 0,$$

which implies $\text{ReLU}(\cdot)$ of a non-positive number is 0. Therefore, whenever $x_i \rightsquigarrow x_j$, we necessarily get $(\bar{p}(W))_{i,j} = 0$.

Conversely, if $\epsilon > \min(\epsilon_0, \epsilon_0^d)$, then it is possible to have
$$\epsilon - \left(\sum_{k=1}^{d} |W|^k\right)_{i,j} > \min(\epsilon_0, \epsilon_0^d) - \min(\epsilon_0, \epsilon_0^d) = 0,$$

which yields $(\bar{p}(W))_{i,j} > 0$. Thus, in this regime ($\epsilon > \min(\epsilon_0, \epsilon_0^d)$), the value $(\bar{p}(W))_{i,j}$ need *not* be zero, even if $x_i \rightsquigarrow x_j$.

Therefore, $(\bar{p}(W))_{i,j} = 0$ is forced (i.e., is *necessary*) by the existence of a directed path $x_i \rightsquigarrow x_j$ if and only if $\epsilon \leq \min(\epsilon_0, \epsilon_0^d)$. This completes the proof. $\qquad\square$

**Proposition 3.** *For $b(W)$ defined in Equation* (16) *and $G(W)$ constructed with edge threshold $\epsilon_0$ in Equation* (14), $(b(W))_{i,j} = 0$ *if there exists at least one directed path from $x_i$ to $x_j$ in $G(W)$, and $(b(W))_{i,j} = 1$ if no such path exists.*

*Proof.* Recall the definition of $b(W)$:

$$b(W) \equiv \mathbb{I}\Big(\sum_{k=1}^{d}\big(\mathbb{I}(|W| \geq \epsilon_0)\big)^k = 0\Big).$$

Let $W^e = \mathbb{I}(|W| \geq \epsilon_0)$ be the *binary* adjacency matrix of $G(W)$, so that

$$W_{i,j}^e \neq 0 \iff (x_i, x_j) \in E(G(W)), \quad \text{and} \quad |W^e| = W^e.$$

Then,

$$(b(W))_{i,j} = 0 \iff \mathbb{I}\Big(\big(p(W^e)\big)_{i,j} = 0\Big) = 0 \iff \big(p(W^e)\big)_{i,j} > 0.$$

By Proposition 2, we know

$$\big(p(W^e)\big)_{i,j} > 0 \iff x_i \rightsquigarrow x_j \in G(W).$$

Hence,

$$(b(W))_{i,j} = 0 \iff x_i \rightsquigarrow x_j \in G(W).$$

Since $b(W)$ takes values in $\{0,1\}^{d \times d}$, we immediately have

$$(b(W))_{i,j} = 1 \iff \neg(x_i \rightsquigarrow x_j) \in G(W).$$

This completes the proof of Proposition 3. $\qquad\square$

**Lemma 3.** *(Necessity) For any $\epsilon$, if there exists at least one directed path from $x_i$ to $x_j$ in $G(W)$ for $G(W)$ constructed by Equation* (14), *then $(\hat{p}(W))_{i,j} = 0$.*

*Proof.* From Proposition 3, we know

$$x_i \rightsquigarrow x_j \in G(W) \iff \big(b(W)\big)_{i,j} = 0.$$

By the definition of $\hat{p}(W)$, whenever $\big(b(W)\big)_{i,j} = 0$, it follows that

$$\big(\hat{p}(W)\big)_{i,j} = 0.$$

Hence,

$$x_i \rightsquigarrow x_j \in G(W) \implies \big(\hat{p}(W)\big)_{i,j} = 0,$$

which completes the proof. $\qquad\square$

**Lemma 4.** *(Sufficiency) For some finite $f(\epsilon_0, \sigma) > \epsilon_0$, if $\epsilon \geq f(\epsilon_0, \sigma)$, then $(\hat{p}(W))_{i,j} = 0$ implies the existence of at least one directed path from $x_i$ to $x_j$ in $G(W)$ constructed with edge threshold $\epsilon_0$ in Equation* (14).

*Proof.* By Lemma 1, there exists a finite threshold $f(\epsilon_0, \sigma) > \epsilon_0$ such that for all $\epsilon > f(\epsilon_0, \sigma)$,

$$(\bar{p}(W))_{i,j} = 0 \implies x_i \rightsquigarrow x_j \in G(W).$$

Next, from Proposition 3, we have

$$(b(W))_{i,j} = 0 \iff x_i \rightsquigarrow x_j \in G(W).$$

By the definition of $\hat{p}(W)$, we know

$$(p(W))_{i,j} = 0 \iff (\bar{p}(W))_{i,j} = 0 \text{ or } (b(W))_{i,j} = 0.$$

Combining these facts, for $\epsilon > f(\epsilon_0, \sigma)$ we get

$$(p(W))_{i,j} = 0 \implies \big(\bar{p}(W)\big)_{i,j} = 0 \text{ or } \big(b(W)\big)_{i,j} = 0 \implies x_i \rightsquigarrow x_j \in G(W).$$

This completes the proof. $\qquad\square$

**Theorem 1.** *There exists at least one directed path from $x_i$ to $x_j$ in $G(W)$ constructed by Equation* (14) *if and only if* $(\hat{p}(W))_{i,j} = 0$, *where $\hat{p}(W)$ is defined by Equation* (16) *with $\epsilon \geq f(\epsilon_0, \sigma)$ for some finite $f(\epsilon_0, \sigma)$.*

*Proof.* Lemma 3 and Lemma 4 together derive this result. □

**Proposition 4.** *The losses $(\hat{p}(W))_{i,j}$ and $\hat{h}(W)$ exhibit a gradient conflict, consistently pushing the parameters of $W$ in opposite directions during optimization:*

$$\forall u, v, \quad \nabla_{W_{u,v}} \hat{h}(W) \cdot \nabla_{W_{u,v}} (\hat{p}(W))_{i,j} \leq 0. \tag{19}$$

*Proof.* Recall $\hat{p}(W) = \bar{p}(W) \circ b(W)$, where $b(W)$ is a binary mask that does not affect the chain rule with respect to $W_{u,v}$. Thus

$$\nabla_{W_{u,v}} (\hat{p}(W))_{i,j} = \begin{cases} \nabla_{W_{u,v}} (\bar{p}(W))_{i,j}, & \text{if } (b(W))_{i,j} = 1, \\ 0, & \text{if } (b(W))_{i,j} = 0. \end{cases}$$

When $(b(W))_{i,j} = 1$ and $(\bar{p}(W))_{i,j} > 0$,

$$\nabla_{W_{u,v}} (\hat{p}(W))_{i,j} = -\nabla_{W_{u,v}} (p(W))_{i,j} = -\sum_{k=1}^{d} \nabla_{W_{u,v}} (|W|^k)_{i,j}.$$

Expanding each derivative over paths that include edge $(x_u, x_v)$ yields

$$\nabla_{W_{u,v}} (|W|^k)_{i,j} = \text{sign}(W_{u,v}) \sum_{p \in \mathcal{P}^{(u,v)}_{i \overset{k}{\rightsquigarrow} j}} \prod_{e \in p \setminus \{(u,v)\}} |W|_e,$$

where $\mathcal{P}^{(u,v)}_{i \overset{k}{\rightsquigarrow} j}$ is the set of all $k$-length directed paths from $x_i$ to $x_j$ that include $(x_u, x_v)$. Since $\prod_e |W|_e \geq 0$, factoring out $\text{sign}(W_{u,v})$ gives

$$\nabla_{W_{u,v}} (\hat{p}(W))_{i,j} = -\text{sign}(W_{u,v}) \sum_{k=1}^{d} \sum_{p \in \mathcal{P}^{(u,v)}_{i \overset{k}{\rightsquigarrow} j}} \prod_{e \in p \setminus \{(u,v)\}} |W|_e.$$

Multiplying by $\text{sign}(W_{u,v})$, we obtain

$$\nabla_{W_{u,v}} (\hat{p}(W))_{i,j} \cdot \text{sign}(W_{u,v}) \leq 0, \tag{38}$$

since the remaining sum is nonnegative.

Next, for

$$\hat{h}(W) = \sum_{k=1}^{d} c_k \left( \sum_{i=1}^{d} |W|^k_{i,i} \right),$$

the gradient w.r.t. $W_{u,v}$ similarly factors out $\text{sign}(W_{u,v})$:

$$\nabla_{W_{u,v}} \hat{h}(W) = \text{sign}(W_{u,v}) \sum_{k=1}^{d} c_k \sum_{i=1}^{d} \sum_{p \in \mathcal{P}^{(u,v)}_{i \overset{k}{\rightsquigarrow} i}} \prod_{e \in p \setminus \{(u,v)\}} |W|_e,$$

where now $\mathcal{P}^{(u,v)}_{i \overset{k}{\rightsquigarrow} i}$ runs over $k$-length loops through $(x_u, x_v)$. Hence, we have:

$$\nabla_{W_{u,v}} \hat{h}(W) \cdot \text{sign}(W_{u,v}) \geq 0, \tag{39}$$

under positive $c_k$ (e.g., all $c_k \geq 0$).

Combining Equations (38) and (39), we have:

$$\nabla_{W_{u,v}} \hat{h}(W) \cdot \nabla_{W_{u,v}} (\hat{p}(W))_{i,j} \leq 0,$$

which completes the proof. □

**Proposition 5.** *Suppose the current graph $G(W)$ contains a directed path from $x_j$ to $x_i$, but there is no path from $x_i$ to $x_j$. In this case, the loss function:*

$$\mathcal{L}(D, W) = F(W; D) + \lambda \|W\| + \gamma (\hat{p}(W))_{i,j} + \rho \hat{h}(W)$$

*prevents the addition of any edge $(u, v)$ that would establish a path from $x_i$ to $x_j$ in an updated graph $G'$, where $G' = (X, E(G) \cup \{(u, v)\})$, as long as $\rho$ is sufficiently large.*

*Proof.* Denote by $p_{j,i}$ any directed path from $x_j$ to $x_i$. We first analyze

$$\nabla_{|W_{u,v}|} \hat{h}(W) = \sum_{q=1}^{d} \sum_{k=1}^{d} \nabla_{|W_{u,v}|} (|W|_{q,q}^{k}).$$

Because each $|W|_{q,q}^{k}$ has nonnegative partial derivatives and $|W_e| \geq \epsilon_0$ for edges present in $G(W)$, we get

$$\nabla_{|W_{u,v}|} \hat{h}(W) \geq \sum_{k=1}^{d} \nabla_{|W_{u,v}|} |W|_{j,j}^{k} \geq \sum_{p_{j,i}, p_{i,u}, p_{v,j}} \epsilon_0^{|p_{j,i}|} \prod_{e \in p_{i,u}} |W_e| \prod_{e \in p_{v,j}} |W_e|,$$

where $p_{j,i}$ ranges over paths $x_j \rightsquigarrow x_i$, $|p_{j,i}|$ denotes the length of the path, $p_{i,u}$ ranges over paths $x_i \rightsquigarrow x_u$ and $p_{v,j}$ ranges over paths $x_v \rightsquigarrow x_j$.

Assume there is *no* directed path $x_i \rightsquigarrow x_j$ in $G(W)$, but adding the edge $(x_u, x_v)$ would create one. Then $G(W)$ must already contain at least one path $x_i \rightsquigarrow x_u$ and one path $x_v \rightsquigarrow x_j$. Thus,

$$\nabla_{|W_{u,v}|} \hat{h}(W) \geq \sum_{p_{j,i}, p_{i,u}, p_{v,j}} \epsilon_0^{|p_{j,i}|} \prod_{e \in p_{i,u}} |W_e| \prod_{e \in p_{v,j}} |W_e| \geq \epsilon_0^{|p_{j,i}| + |p_{i,u}| + |p_{v,j}|}.$$

Denote $k_0 = |p_{j,i}| + |p_{i,u}| + |p_{v,j}|$.

Next, consider the overall gradient of

$$\mathcal{L}(D, W) = F(W; D) + \lambda + \gamma (\hat{p}(W))_{i,j} + \rho \hat{h}(W).$$

Taking the partial w.r.t. $|W_{u,v}|$,

$$\nabla_{|W_{u,v}|} \mathcal{L}(D, W) = \underbrace{\nabla_{|W_{u,v}|} F(W; D) + \lambda + \gamma \nabla_{|W_{u,v}|} (\hat{p}(W))_{i,j}}_{\equiv \alpha} + \rho \nabla_{|W_{u,v}|} \hat{h}(W).$$

Since $\nabla_{|W_{u,v}|} \hat{h}(W) \geq \epsilon_0^{k_0}$, we have

$$\nabla_{|W_{u,v}|} \mathcal{L}(D, W) \geq \alpha + \rho \epsilon_0^{k_0}.$$

Hence, if

$$\rho \geq \frac{|\alpha|}{\epsilon_0^{k_0}},$$

then $\nabla_{|W_{u,v}|} \mathcal{L}(D, W) \geq 0$, forcing the gradient update to *reduce* $|W_{u,v}|$ (i.e., disfavor adding the edge $(x_u, x_v)$).

Thus, for all $\rho$ above that finite threshold $|\alpha|/\epsilon_0^{k_0}$, the absolute weight of $(x_u, x_v)$ is pushed down to forbid its presence in $G(W)$. This completes the proof. $\square$

**Proposition 6** ((Ban et al., 2024, Proposition 2)). *Given a partial order set $O = \{x_i \prec x_j\}$, a DAG $G$ satisfies $O$ if and only if, for all $x_i \prec x_j$ in $O^+$, there exists no directed path from $x_j$ to $x_i$ in $G$.*

*Proof.* We aim to establish the equivalence:

There exists at least one topological order $\pi$ of $G$ that satisfies $O \iff \forall\, x_i \prec x_j \in O^+,\ \neg(x_j \rightsquigarrow x_i)$ in $G$.

($\Longrightarrow$): Assume there is a topological ordering $\pi$ of $G$ that satisfies $O$. By definition of topological ordering, if $x_i$ is an ancestor of $x_j$ in $G$, then $x_i$ must appear before $x_j$ in $\pi$. Because $\pi$ respects the constraints in $O$, for every $(i, j) \in O^+$ we have $x_i$ placed before $x_j$ in $\pi$. Suppose, for contradiction, that there is a directed path $x_j \rightsquigarrow x_i$ in $G$. Then $x_j$ is an ancestor of $x_i$ and should appear before $x_i$ in *any* topological ordering, contradicting the fact that $x_i$ is before $x_j$ in $\pi$. Hence, no such path $x_j \rightsquigarrow x_i$ exists, proving this direction.

($\Longleftarrow$): Conversely, suppose that for every $(i, j) \in O^+$, there is no directed path $x_j \rightsquigarrow x_i$ in $G$. Since $G$ is a DAG, it possesses at least one topological ordering (constructible via, e.g., Kahn's algorithm or a depth-first search approach). We now argue that this ordering can be adjusted, or equivalently, we can choose a valid ordering outright, so that all pairs in $O$ are respected. Because the absence of a path $x_j \rightsquigarrow x_i$ ensures no cycle is created by demanding $x_i$ precede $x_j$, none of the constraints in $O^+$ is violated. Thus we can finalize a topological ordering $\pi$ of $G$ in which, for each $(i, j) \in O^+$, $x_i$ appears before $x_j$. Hence $G$ has a topological ordering that satisfies all order constraints in $O$.

Combining both directions, we conclude that the existence of a topological ordering of $G$ satisfying $O$ is equivalent to having no path $x_j \rightsquigarrow x_i$ in $G$ for every $(i, j) \in O^+$. $\qquad\square$

**Proposition 7.** *Each optimal solution of the equality-constrained problem* (5) *corresponds to the global optimum of the following convex optimization problem:*

$$\min_{W \in \mathbb{R}^{d \times d}} \|D(W \circ M) - D\|_2^\ell + \lambda \|W\|_1, \tag{25}$$

*where $M_{i,j} = 1$ if $\pi(i) < \pi(j)$ and $M_{i,j} = 0$ otherwise, for a specific permutation $\pi$ of $\{1, 2, \ldots, d\}$.*

*Proof.* We begin with the equality-constrained problem

$$\min_{W \in \mathbb{R}^{d \times d}} \|D\,W - D\|_2^\ell + \lambda \|W\|_1, \quad \text{subject to} \quad h(W) = 0. \tag{40}$$

Let $W^*$ be an optimal solution to (40). By feasibility, $h(W^*) = 0$, and by the stationarity condition (assuming a Lagrangian approach),

$$\nabla f(W^*) + \mu \nabla h(W^*) = 0, \tag{41}$$

where $f(W) = \|D\,W - D\|_2^\ell + \lambda \|W\|_1$ and $\mu$ is the Lagrange multiplier associated with $h(W) = 0$.

By definition of $h(\cdot)$, the condition $h(W^*) = 0$ means that $G(W)$ is DAG, and there is a permutation $\pi$ of $\{1, 2, \ldots, d\}$ such that

$$W_{i,j}^* = 0 \quad \text{whenever} \quad \pi(i) \geq \pi(j).$$

Equivalently, $W^*$ is *strictly upper-triangular* up to reindexing of rows and columns by $\pi$. Define a binary matrix $M \in \{0, 1\}^{d \times d}$ by

$$M_{i,j} = \begin{cases} 1, & \text{if } \pi(i) < \pi(j), \\ 0, & \text{otherwise.} \end{cases}$$

Then $W^* \circ M = W^*$ because multiplying by $M_{i,j}$ zeroes out all entries below or on the "$\pi$-diagonal."

Consider the convex program

$$\min_{W \in \mathbb{R}^{d \times d}} \left\| D(W \circ M) - D \right\|_2^\ell + \lambda \|W\|_1. \tag{42}$$

Since $W \circ M$ zeroes out the same pattern enforced by $M$, the feasible set effectively restricts $(W \circ M)$ to that strict upper-triangular shape under $\pi$. Let $W'$ be an optimal solution to (42); by standard convexity arguments, stationarity implies

$$\nabla \left\| D(W' \circ M) - D \right\|_2^\ell + \lambda \nabla \|W'\|_1 = 0. \tag{43}$$

(This is taken with respect to $W'$, understanding that the chain rule also includes the elementwise product with $M$, but crucially $(W' \circ M)$ is valid for all nonzero coordinates consistent with the triangular structure.)

We claim $W^*$ also satisfies the stationarity for (42). Note first that

$$W^* \circ M \;=\; W^*.$$

The equality $h(W^*) = 0$ (strict upper-triangular structure) also implies that any partial derivative of $h(\cdot)$ w.r.t. $W_{i,j}^*$ is zero whenever $\pi(i) < \pi(j)$. Concretely, if we treat $h(\cdot)$ as a sum of path-indicator terms, no path $q \rightsquigarrow i$ and $j \rightsquigarrow q$ can coexist unless $\pi(j) < \pi(q) < \pi(i)$, contradicting $\pi(i) < \pi(j)$. Hence:

$$\nabla_{W_{i,j}} h\big(W^*\big) \;=\; 0 \quad \text{for } \; \pi(i) < \pi(j).$$

Because those are precisely the coordinates in which $W^*$ could be nonzero, we get

$$\nabla h\big(W^* \circ M\big) \;=\; 0.$$

Returning to (41),

$$\nabla f\big(W^* \circ M\big) \;+\; \mu \,\nabla h\big(W^* \circ M\big) \;=\; 0 \;\implies\; \nabla f\big(W^* \circ M\big) \;=\; 0.$$

Thus $W^*$ satisfies the first-order condition for problem (42), i.e.

$$\nabla \Big\| D\big(W^* \circ M\big) - D \Big\|_2^{\ell} \;+\; \lambda \,\nabla \|W^*\|_1 \;=\; 0.$$

Since (42) is a convex optimization problem (sum of a smooth convex term and an $\ell_1$-regularization), any $W$ that satisfies the stationarity condition must be a global optimum. From (43) and the stationarity for $W^*$, we see $W'$ and $W^*$ both satisfy the same gradient condition. Therefore $W^* = W'$.

We have shown that every global optimum $W^*$ of (40) (with $h(W^*) = 0$) coincides with the global optimum $W'$ of the convex problem (42) defined by the matrix $M$ associated with the same permutation $\pi$. This completes the proof. $\square$

## D. Complete Experimental Results and Analysis

This section provides a comprehensive analysis of our experimental results. First, we evaluate additional aspects not covered in the main text, including time complexity, results on nonlinear data, the integration of path absence constraints, the results of various backbone algorithms and the impact of varying constraint numbers. Then, we present supplementary results under more extensive settings for the experiments discussed in the main text.

### D.1. Time Complexity

The primary computational cost in our approach arises from matrix operations. The time complexity of a standard matrix multiplication for a $d \times d$ matrix is $O(d^3)$. The key function in our proposed loss, $\hat{p}(W)$, involves computing $p(W) = \sum_{k=1}^{d} |W|^k$, which has a time complexity of $O(d^4)$. This exceeds the computational cost of commonly used acyclicity losses, which are $O(d^3)$. However, this additional overhead can be mitigated through GPU acceleration.

To evaluate the efficiency of our method, we implemented two versions: one using a CPU for all computations and another utilizing a GPU for computing the gradients of the newly introduced loss terms. The data approximation loss and acyclicity loss were computed on the CPU in both cases.

We denote the process of solving the partial order-constrained problem as **PO-NOTEARS** and the process of using the PO-NOTEARS result as an initial guess for solving the path existence-constrained problem as **PE-NOTEARS-path**. Note that the sum of run time of **PO-NOTEARS** and **PE-NOTEARS-path** is the total time consumed by **PE-NOTEARS**, the method used to integrate ancestral constraints.

Figure 4 compares the runtime of these approaches using both CPU and GPU implementations. The results indicate that solving the PO-constrained problem is the most computationally expensive step, whereas solving the PE-constrained problem is significantly faster, with a runtime comparable to the standard NOTEARS algorithm.

Additionally, GPU acceleration provides a large speedup, particularly as the number of nodes increases. This demonstrates the potential for integrating our method into deep learning systems, where efficient GPU-based implementations are crucial.

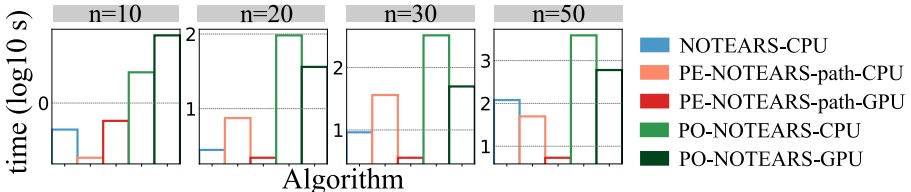

*Figure 4.* Comparison time between GPU and CPU versions of the proposed method.

## D.2. Improved Time Complexity by Accelerating Matrix Power Iteration

Zhang et al. (2022) introduced an fast truncated matrix power iteration (fast TMPI) algorithm that is suitable for accelerating the calculation of matrix power iteration defined in the path existence loss, which is in complexity $O(d^3 \log d)$, comparable with the complexity $O(d^3)$ of common acyclicity losses.

Below, we compare fast TMPI against the original direct matrix power operation under the following setting: ER2 graph, linear SEM with Gaussian noise, and 80% prior paths. Results of the run time (in seconds) and F1 score are reported as **Direct Matrix Power / fast TMPI** for PE-NOTEARS under varioud node numbers (@Node).

| @Node | 10 | 20 | 30 | 50 |
|---|---|---|---|---|
| Time (s) | 29.8 / 26.7 | 106.4 / 90.3 | 269.3 / 188.3 | 973.6 / 500.8 |
| F1 Score | 0.87 / 0.88 | 0.77 / 0.77 | 0.68 / 0.68 | 0.61 / 0.61 |

We observe that PE-NOTEARS with fast TMPI achieves a significant speedup on large-scale graphs while maintaining comparable performance. This confirms that fast TMPI integrates effectively into our approach, reducing time complexity from $O(d^4)$ to $O(d^3 \log d)$, making it competitive with the $O(d^3)$ complexity of NOTEARS.

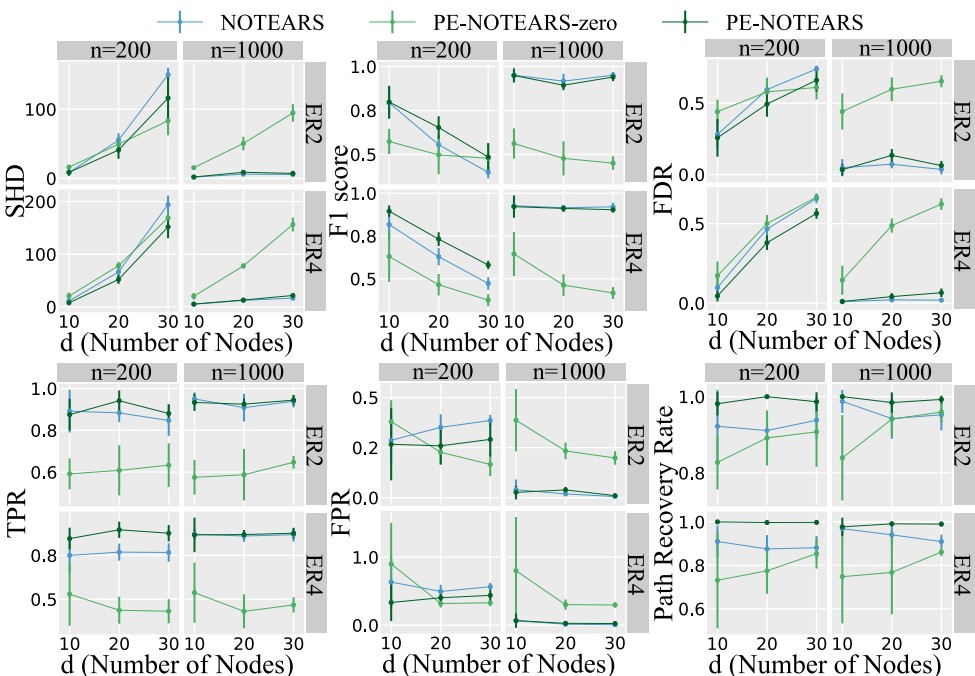

*Figure 5.* Comparison and ablation results on nonlinear data.

## D.3. Results on Nonlinear Data

This experiment evaluates our method on nonlinear data using NOTEARS-MLP (Zheng et al., 2020) as the backbone algorithm. The nonlinear data is generated based on the binary mask of the ground truth structure $W$. Uniformly random weights are assigned to the elements where $W_{i,j} = 1$, denoted as $W_1, W_2, W_3$. Given these weights, samples are generated according to the following nonlinear model:

$$D = \tanh(DW_1) + \cos(DW_2) + \sin(DW_3) + z, \quad z \sim \mathcal{N}(0, 1). \tag{44}$$

We compare the performance of the NOTEARS, PE-NOTEARS, and PE-NOTEARS-zero (which removes order-guided optimization and uses a zero matrix as the initial guess), using MLP to fit data distributions (Zheng et al., 2020). The results, reported in Figure 5, show that PE-NOTEARS consistently outperforms both NOTEARS and PE-NOTEARS-zero, achieving lower SHD, higher F1-score, and lower FDR.

Additionally, PE-NOTEARS demonstrates a significantly higher TPR compared to NOTEARS, indicating that incorporating path existence constraints effectively enhances the recovery of causal links. PE-NOTEARS-zero achieves higher TPR than PE-NOTEARS but at the cost of recovering a large number of redundant edges, as reflected in its high SHD and FDR.

Regarding the path recovery rate, PE-NOTEARS consistently recovers the most paths from priors, confirming the effectiveness of order-guided optimization in resolving prior violations. Notably, despite PE-NOTEARS-zero introducing many redundant edges, it still fails to match PE-NOTEARS in satisfying path existence constraints. This further validates the importance of using a partial order-guided initial guess to support path existence constraints.

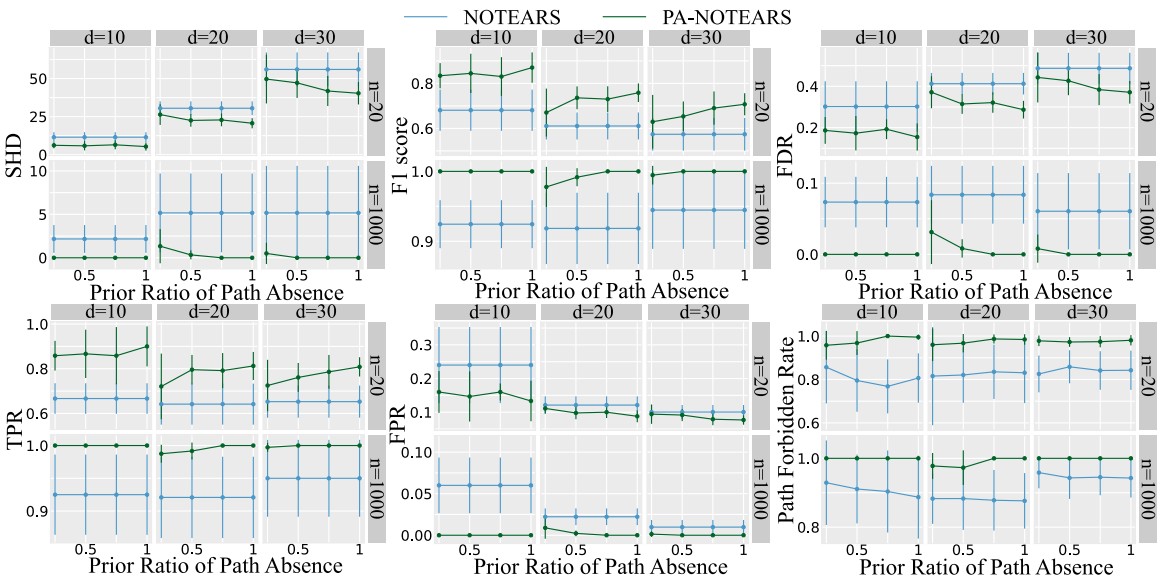

*Figure 6.* Comparison results of the path absence-constrained to data-based structure learning.

## D.4. Results on Path Absence Constraints

This experiment evaluates structure learning with path absence constraints. Since the characterization of path absence is straightforward, we did not include these results in the main text.

Using the characterization $(p(W))_{i,j} = 0$, where $p(W)$ is defined as:

$$p(W) \equiv \sum_{k=1}^{d} |W|^k. \tag{45}$$

This condition is equivalent to enforcing the absence of a directed path $x_i \rightsquigarrow x_j$.

Given a path absence mask $B \in \{0,1\}^{d \times d}$, where $B_{i,j} = 1$ represents the constraint $\neg(x_i \rightsquigarrow x_j)$ in $G(W)$, we formulate the structure learning task as:

$$\min_{W \in \mathbb{R}^{d \times d}} \|DW - D\|_2^{\ell} + \lambda \|W\|_1 + \beta \cdot \sum (p(W) \circ B) \quad \text{s.t. } h(W) = 0. \tag{46}$$

The path absence loss weight is set to $\beta = 1$. We denote the method solving this task as PA-NOTEARS and compare it with standard NOTEARS (without any prior constraints). The comparison results, with varying proportions of path absence constraints on ER-2 graphs, are shown in Figure 6. The results indicate that PA-NOTEARS consistently outperforms standard NOTEARS across all metrics, with greater improvements observed as the number of prior constraints increases. This improvement is expected, as path absence constraints are inherently aligned with the acyclicity loss in gradient direction, making them easy to enforce. The remaining unsatisfied constraints in the experiments are primarily due to the trade-off between data approximation and path absence loss.

**Discussion**    Notably, path absence constraints can be viewed as an extension of the acyclicity constraint that further reduces the search space for optimal solutions. This connection aligns closely with the partial order constraints discussed in the main text, reinforcing the effectiveness of integrating path absence information into differentiable structure learning.

**Comparison with Path Existence**    A complete specification of path absence constraints effectively imposes a precise restriction on the set of potential edges in the graph. Specifically, we consider edge absence constraints as a subset of path absence constraints, since the absence of a path $x_i \rightsquigarrow x_j$ implies the absence of a direct edge $x_i \rightarrow x_j$. Thus, fully specifying all path absence constraints narrows the possible edge set to only those in the ground truth. This makes path absence constraints strictly stronger than total ordering constraints, as a total order defines the set of potential edges (i.e., those respecting the order), while path absence constraints further eliminate incorrect edges.

In contrast, path existence constraints do not offer the same level of restriction. Even when all path existence constraints are specified, they do not uniquely determine any particular edge without additional assumptions. Moreover, path existence constraints provide no information about edge absence, so the overall space of potential edges remains large. As a result, we observe highly accurate outputs when many path absence constraints are available. However, such comprehensive prior knowledge is rarely accessible in practice. For $d$ variables, specifying $O(d^2)$ path absence constraints is required to achieve this level of restriction, whereas significant improvements can be achieved with only $O(d)$ path existence or partial order constraints. For example, a total order requires just $d - 1$ partial orders or path existence constraints.

This comparison underscores the practical advantage of path existence constraints: while path absence provides stronger theoretical restriction, path existence is far more scalable and feasible in real-world applications.

### D.5. Results with Varying Numbers of Path Existence Constraints

This experiment examines the impact of different numbers of path existence constraints on structure learning performance. We vary the prior percentage $q$ and evaluate its influence on several PE-constrained approaches, including PE-NOTEARS-intuitive (using $\bar{p}$ as the path existence loss), PE-NOTEARS-zero (using zero initialization), and PE-NOTEARS (ours). The results on ER-2 graphs are presented in Figure 7.

The findings indicate that PE-NOTEARS consistently outperforms other methods across different numbers of path constraints. An exception occurs in cases with 10 nodes, where PE-NOTEARS-zero performs comparably to PE-NOTEARS. This is because, with only 10 nodes, the ordering space is relatively simple, reducing the severity of order-violating issues even without order-guided initialization.

As the number of prior paths increases, the performance of PE-NOTEARS further improves. Notably, despite incorporating more prior path constraints, PE-NOTEARS is still accurate in edge recovery (as reflected in stable FDR and FPR values). In contrast, both ablation variants exhibit a substantial increase in false discoveries with more constraints.

These results further validate the effectiveness of using $\hat{p}$ as the path existence function and employing partial order-guided optimization strategies for path existence-constrained structure learning.

### D.6. Results on Various Backbone Algorithms

This section evaluates the effectiveness of integrating path existence constraints into different backbone differentiable structure learning algorithms using our approach. The results for DAGMA and GOLEM are presented in Figures 8 and 9,

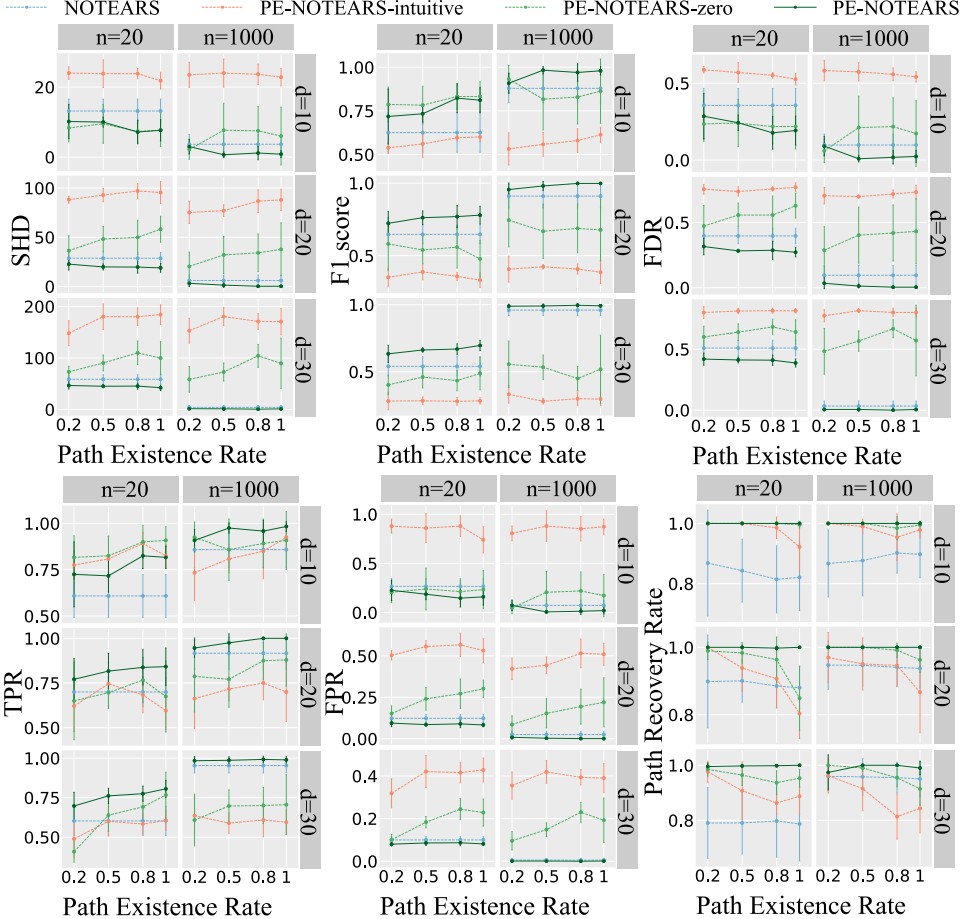

*Figure 7.* Comparison and ablation results with varying path existence constraint numbers.

respectively.

The findings indicate that our method significantly enhances the performance of both DAGMA and GOLEM by effectively incorporating path existence constraints. Additionally, the high path recovery rate demonstrates the effectiveness of order-guided optimization in resolving conflicts among prior constraints across different approaches.

These results confirm the broad applicability of our method, showing that it generalizes well to various differentiable structure learning frameworks for integrating path existence constraints.

### D.7. Supplementary Results and Analysis from the Main Text

This section presents the complete experimental results corresponding to the settings discussed in the main text. The supplementary results are shown in Figures 10, 11, 12, 13, 14, and 15.

The specific experimental settings for these supplementary results are detailed in the captions of the respective figures.

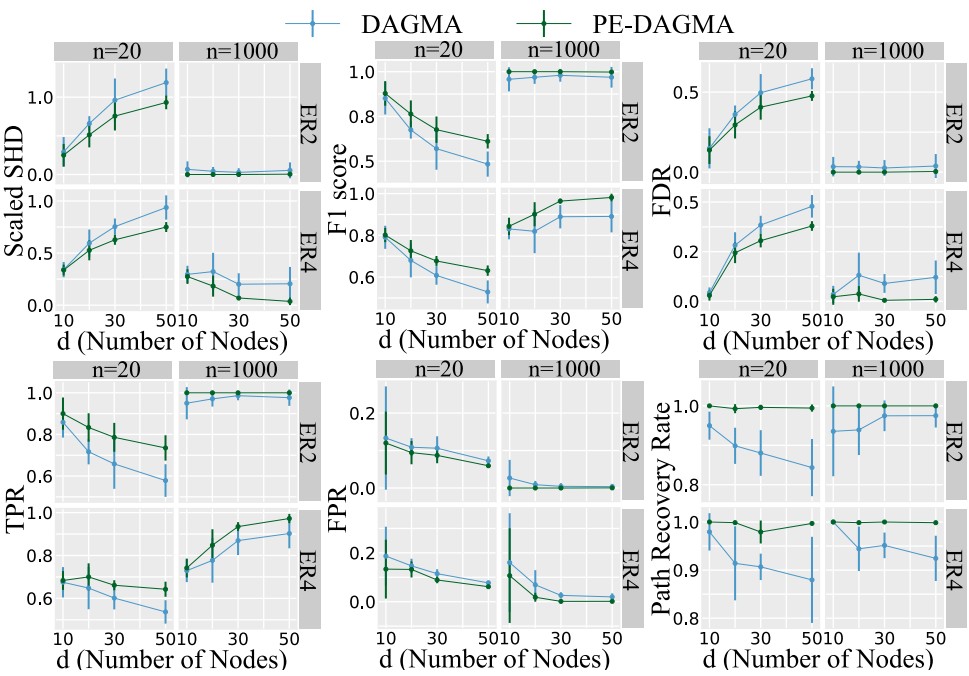

*Figure 8.* Results of our approach with DAGMA.

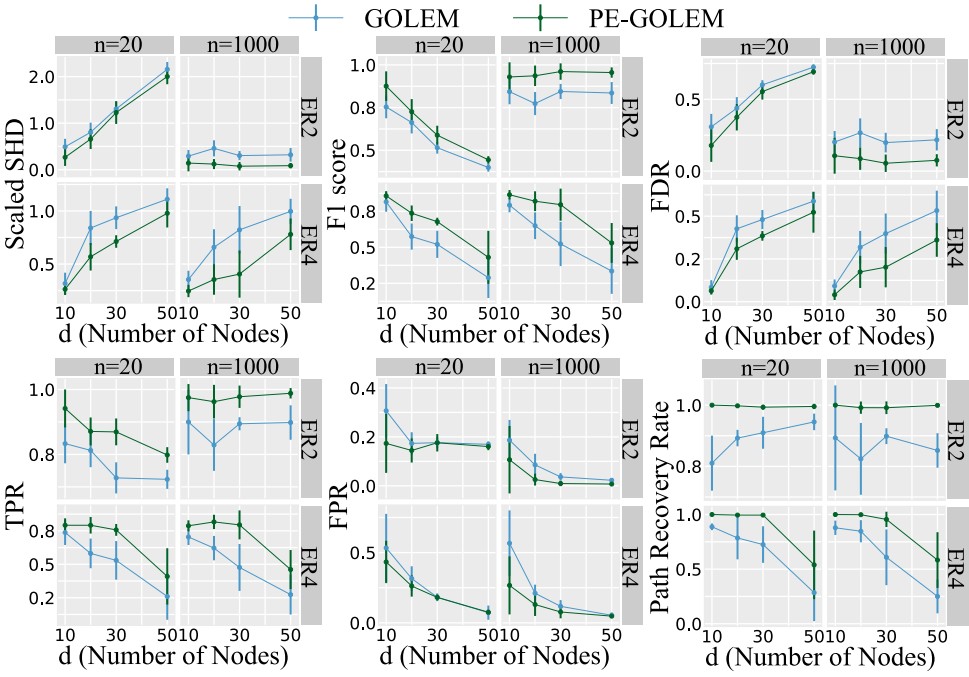

*Figure 9.* Results of our approach with GOLEM.

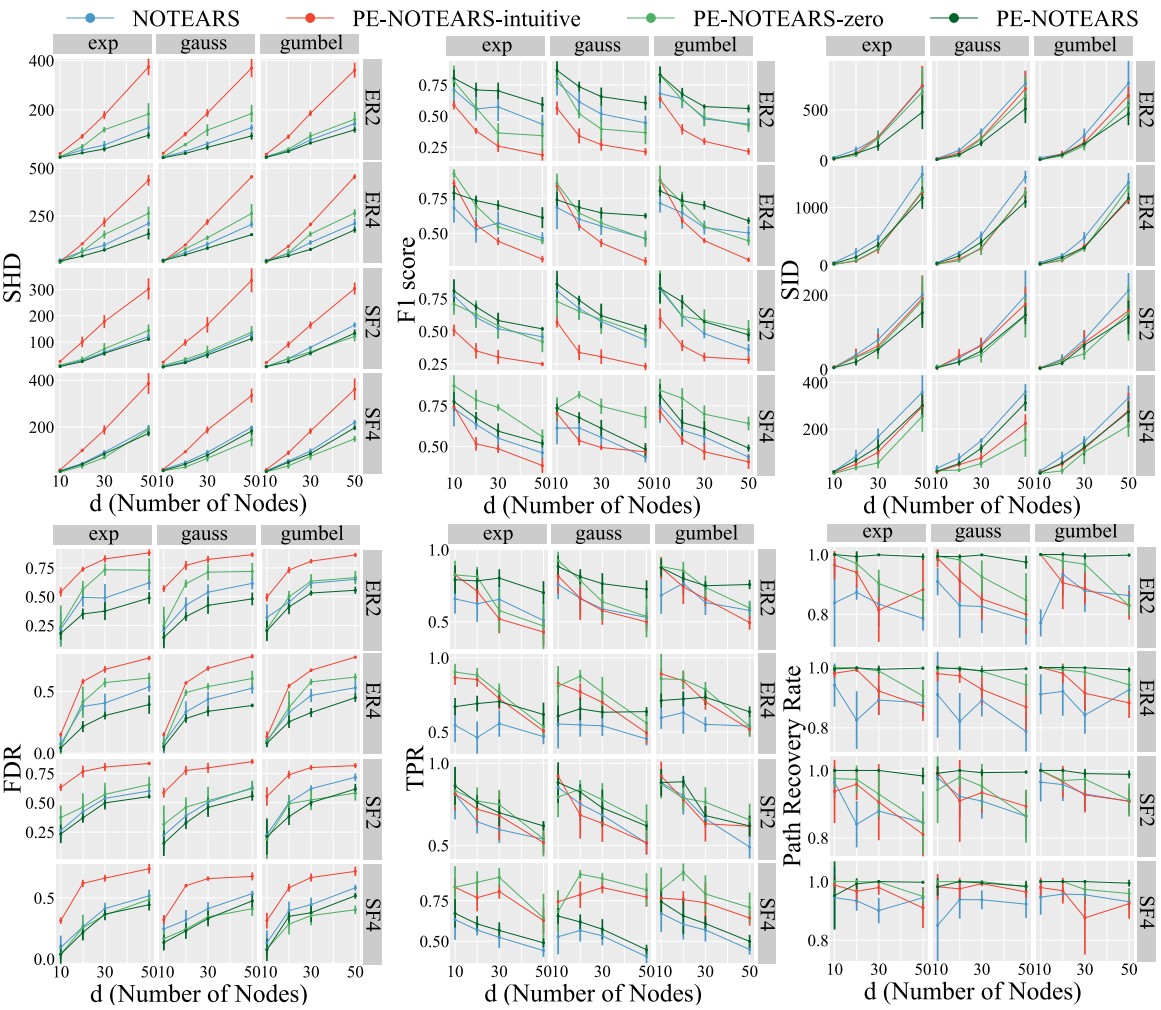

*Figure 10.* Supplementary comparison and ablations results with sample size $n = 20$.

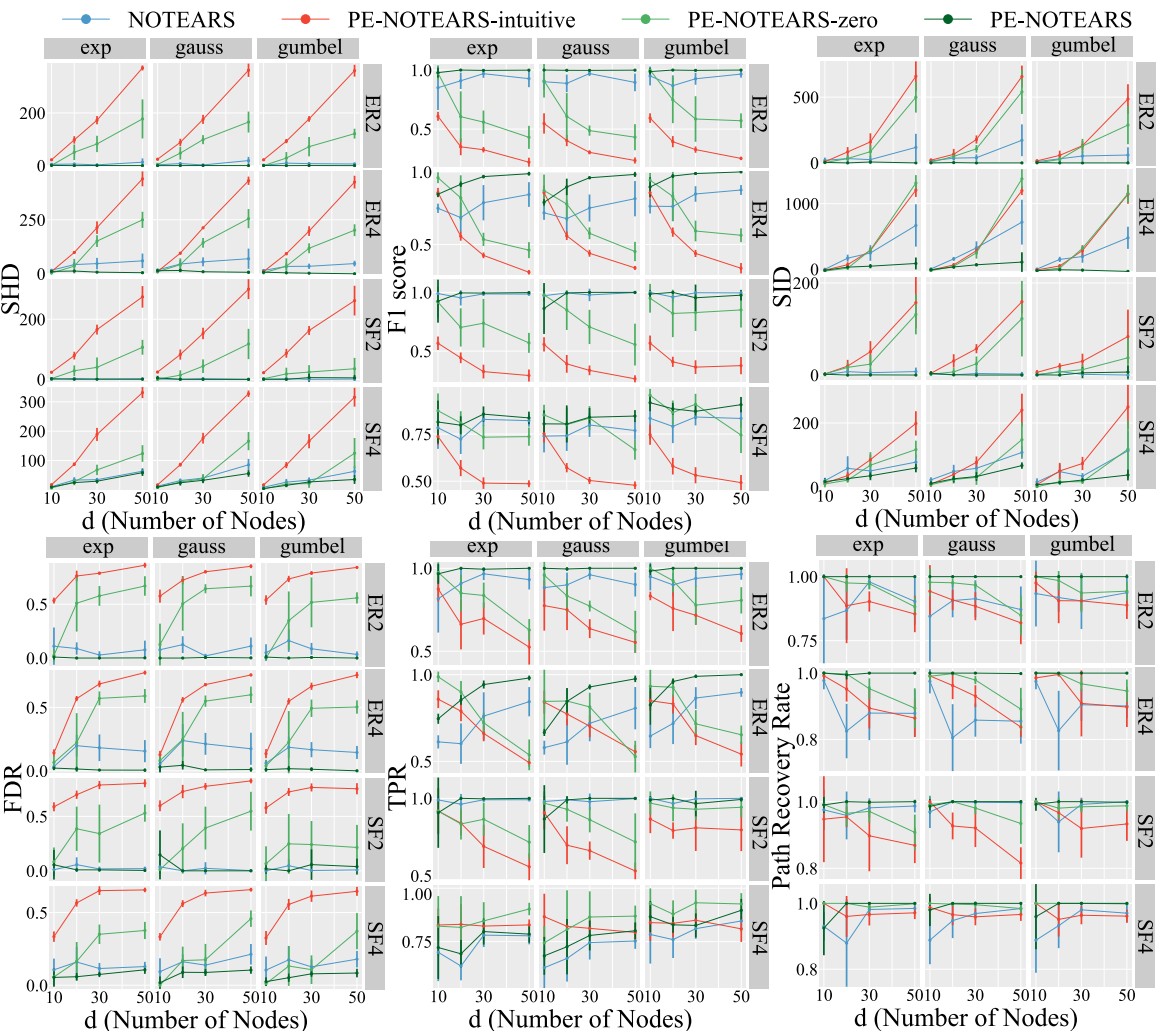

*Figure 11.* Supplementary comparison and ablations results with sample size $n = 1000$.

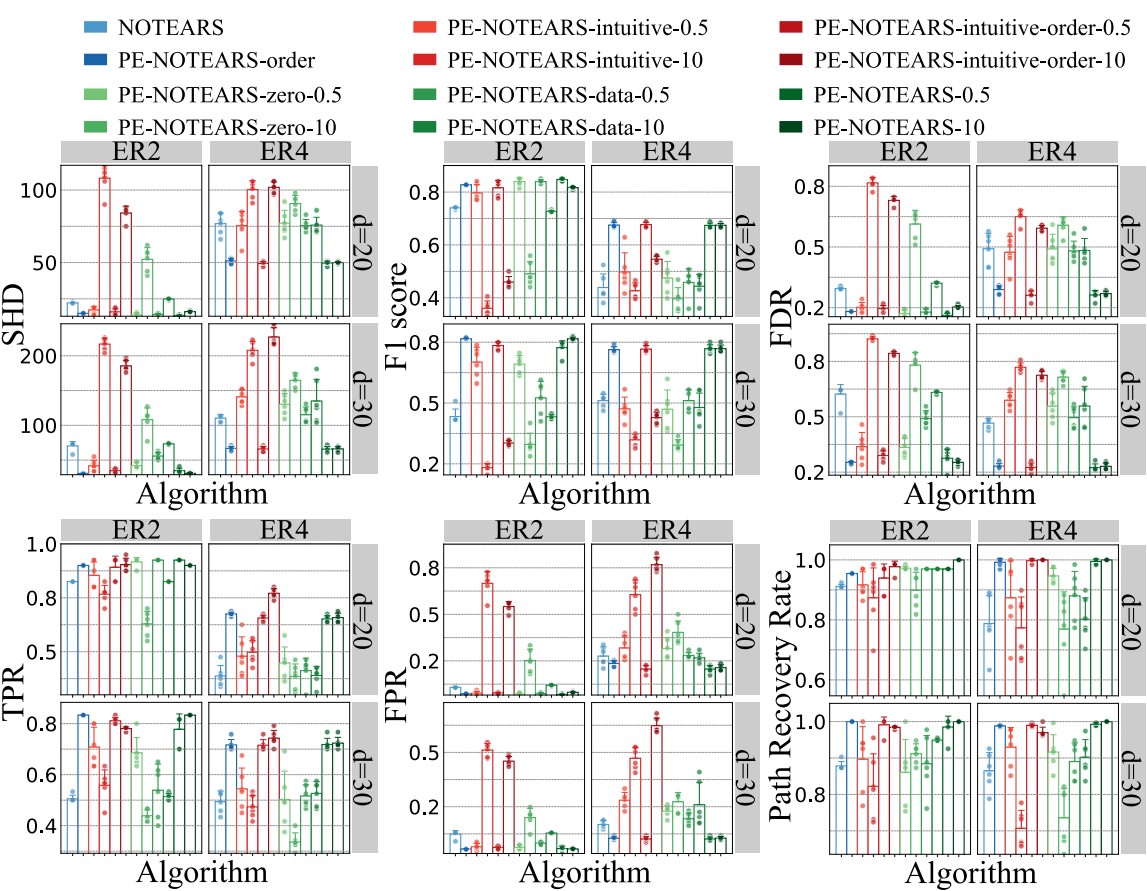

*Figure 12.* Supplementary ablation results on initial random guesses with sample size $n = 20$. The suffix $0.5$ and $10$ in the method reference is the value of path existence threshold $\epsilon$.

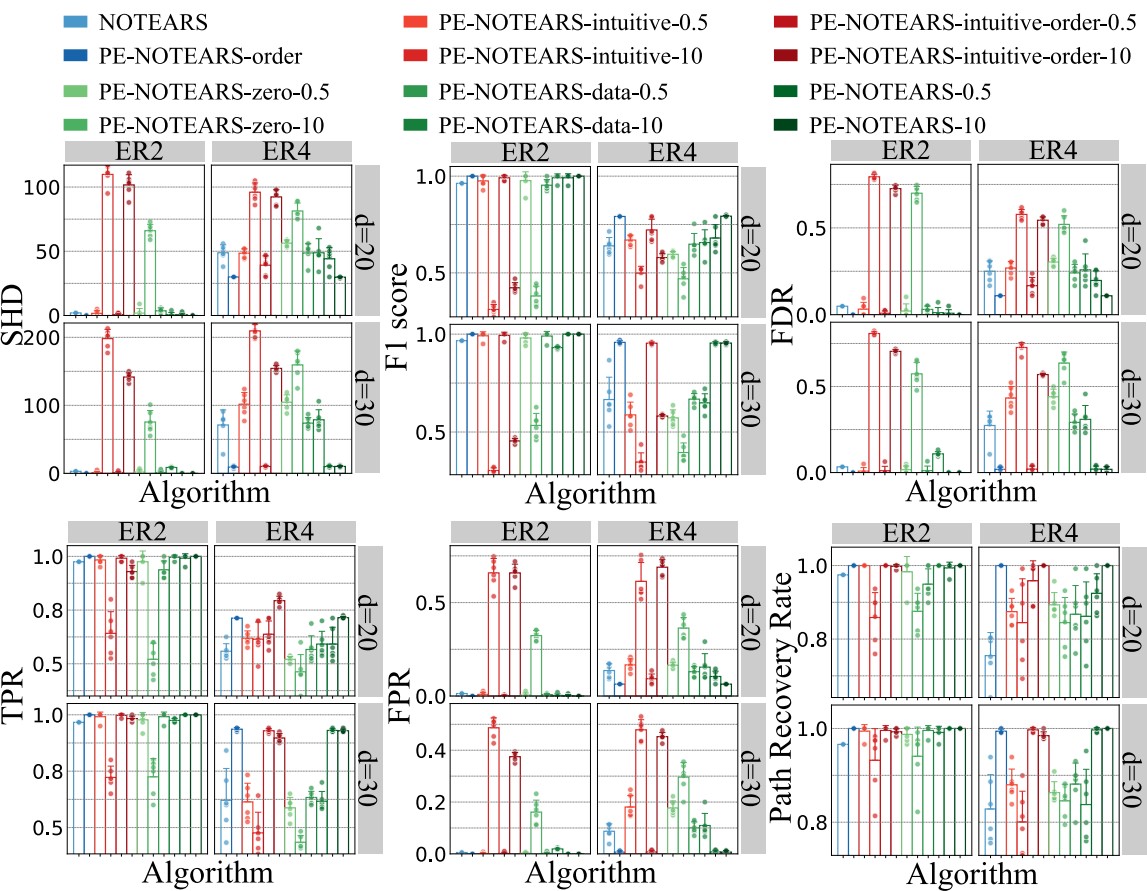

*Figure 13.* Supplementary ablation results on initial random guesses with sample size $n = 1000$. The suffix $0.5$ and $10$ in the method reference is the value of path existence threshold $\epsilon$.

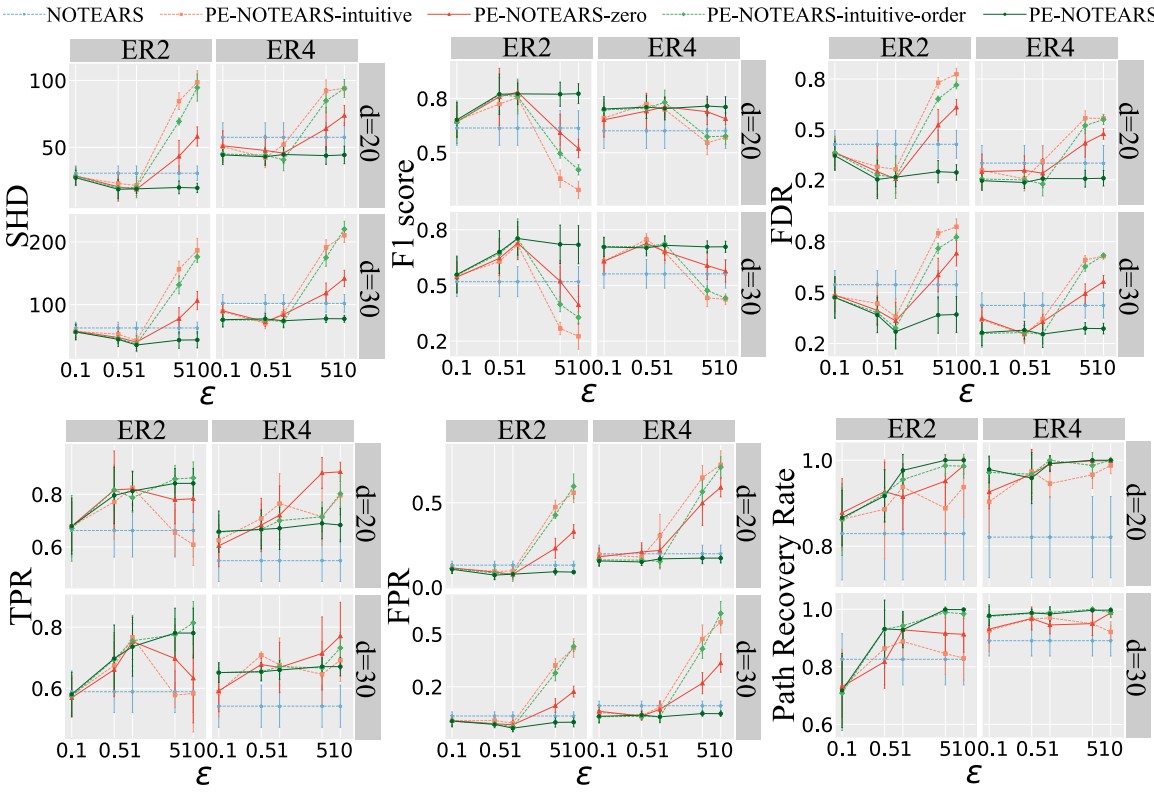

*Figure 14.* Supplementary ablation results with varying path thresholds with sample size $n = 20$. PE-NOTEARS-intuitive-order refers to the method using $\bar{p}$ as path existence loss and the partial order-guided initial guess.

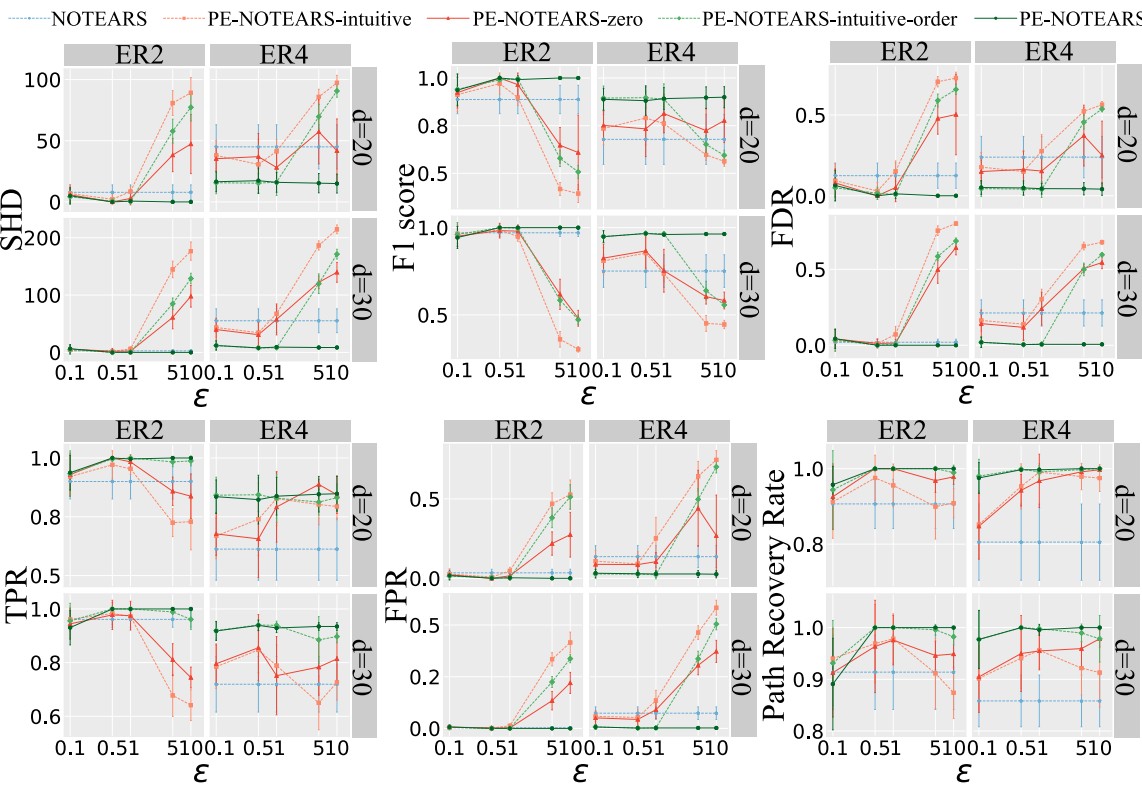

*Figure 15.* Supplementary ablation results with varying path thresholds with sample size $n = 1000$. PE-NOTEARS-intuitive-order refers to the method using $\bar{p}$ as path existence loss and the partial order-guided initial guess.

