# OpenReview forum: "Differentiable Structure Learning with Ancestral Constraints"
_ICML.cc/2025/Conference — ICML 2025 poster_

### Official Review · Reviewer_MDE7 · 2025-02-23

**Overall Recommendation:** 3

**Summary:**

The paper proposes a method to integrate prior knowledge of presence/absence of certain edges or paths into differentiable structural learning frameworks. The paper presents theoretical analyses of several strategies, and the related issues, for handling such constraints in a continuous optimization regime. The authors then propose an alternative continuous characterization of the constraints along with an order-guided optimization strategy to mitigate the identified issues.

### Update after rebuttal:
I retain my view regarding the novelty of the practical algorithm of the method, which to me is a incremental, rather limited, extension of the existing frameworks (those mentioned in the discussions). I understand that the other reviewers may hold a different viewpoint regarding this. However, I highly value the theoretical analysis of the problem at hand and I think it can contribute to the understanding of optimization behavior induced by order constraints.

After reading the authors' comment, I am slightly more positive about the work and decided to raise my score to 3, though honestly a score of 2.5 would better justify the update. This is also to encourage the authors to clarify the similarities/differences of the approach compared to existing methods, if the authors in fact leverage their results.

**Claims And Evidence:**

The paper provides solid theoretical results for most of the key claims and extensive empirical evidence overall to substantiate the effectiveness of the proposed method. However, the following points are problematic to me.

1. In Section 4.1, the authors point out an issue of violating paths when directly optimize Eq. (18) due to the sensitivity to different intializations for $W$. In the original implementation of NOTEARS or DAGMA, which the authors are also using, $W$ is often initialized at zeros and this has so far been shown to work sufficiently well in many settings. Furthermore, while the case in Proposition 5 may arise, a positive view from Proposition 5 is that if the current graph contains correct paths, the loss will help prevent edges that causes conflicting paths. To fully establish it as a motivation for the proposed method, the authors should provide some empirical results for the prevalence of such issue in practice, which is currently lacking in the manuscript.

2. The claim this is the first work that addresses the integration of ancestral constraints into differentiable structure learning is rather misleading. At least based on the authors' discussions of related works in Appendix A, we have works with same problem setup like Wang et al. (2024), Sun et al. (2023), Ban et al. (2024) (where ancestral constraints are part of the partial order constraints).

**Essential References Not Discussed:**

As for order-based constrained optimization, Deng et al. (2024) is a recent work that improves NOTEARS by leveraging prior knowledge of topological ordering. The authors can consider this method as an alternative for the currently proposed one in step 2.

*Deng, C., Bello, K., Aragam, B., & Ravikumar, P. K. (2023, July). Optimizing notears objectives via topological swaps. In International Conference on Machine Learning (pp. 7563-7595). PMLR.*

**Experimental Designs Or Analyses:**

The authors conduct experiments on well-established benchmarks and report the standard metrics for evaluation.

My main concern is in the lack of baseline comparison and that the paper only reports the performance of the proposed method. It is not surprising that incorporating additional order-based constraints yield performance improvement, which is expected. As mentioned above, there are other algorithms for integrating constraints, certainly with different specifications and not necessarily fully differentiable (e.g., O’Donnell et al., 2006, Chen et al., 2016).  Therefore, it is important to have an empirical assessment of these methods as baselines to understand how well the proposed method works. In particular, the authors should compare the method with Wang et al. (2024), Sun et al. (2023) to empirical verify the sub-optimality of the ReLU characterization.

**Methods And Evaluation Criteria:**

My understanding from Section 4.2 is that the proposed method involves two steps: (1) solve for $W$ with path absence constraints by Eq (23), (2) use the resulting $W$ as initial points and solve for final graph by Eq (25) with path presence constraints.

From Section 4.2, the task is reformulated into an structure learning objective with partial order constraints, which Ban et al. (2024) have already addressed, also stated so by the authors in lines 736 - 738. Furthermore, Eq. (25) corresponds to the optimization problem with total ordering constraint and it is the relaxation of such constrained optimization problem proposed in Eq. (7) of Ban et al. Therefore, Proposition 7 seems to me purely a re-statement of Proposition 3 of Ban et al. (2024). Furthermore, the insight from Proposition 7 is not new in that acyclicity constraint is no longer needed given the knowledge of total ordering, which has been actively exploited; some recent works include Deng et al., (2023), Shahverdikondori et al. (2024)

*Deng, C., Bello, K., Aragam, B., & Ravikumar, P. K. (2023, July). Optimizing notears objectives via topological swaps. In International Conference on Machine Learning (pp. 7563-7595). PMLR.*

*Shahverdikondori, M., Mokhtarian, E., and Kiyavash, N.QWO: Speeding up permutation-based causal discovery in liGAMs. In The Thirty-eighth Annual Conference on Neural Information Processing Systems, 2024*

**Other Comments Or Suggestions:**

I currently vote for a weak reject mainly due to the questionable novelty of the proposed theoretical results in relation to the existing works. However, I am willing to raise the score if the authors could provide explanations as well as additional empirical evidence to demonstrate the competitiveness or superiority of the proposed method. Some further comments are:

1. In line 734, the ReLU constraints are discussed in works like Wang et al. (2024) and Sun et al. (2024), which also need to be explicitly cited in the Section 3.1 discussion.

2. Furthermore, I do not find the implementation of DAGMA in the provided code. Despite the filename, the codes have NOTEARS implementation.

3. The authors should summarize the final algorithm to highlight what the key contribution is.

**Other Strengths And Weaknesses:**

**Strengths:**

The paper offers a systematic analysis of the challenges associated with incorporating ancestral constraints into differentiable causal discovery frameworks, both from a theoretical and empirical perspective. In my view, this is the key point of differentiation of this work compared to previous studies. While the concept of applying binary masking in causal discovery is not new, the in-depth theoretical analysis presented here is particularly interesting. Given the maturity of the gradient-based causal discovery literature, it is crucial to explore the optimization behavior and the role that different constraints play in the optimization process.

**Weaknesses:**

As mentioned above, the proposed objective to me bears striking resemblances with the previous results, particularly Ban et al. (2024). Apart from those, the result in Eq (16) is also highly relevant. The proposed binary-masked characterization is a threshold-dependent relaxed equivalence of the characterization in Proposition 4 of Ban et al. which states that no directed path from $X_i$ to $X_j$ if and only $(\sum_{k=1}^{d} (W \circ W)_{i,j}=0$. Here $|W|$ and $W \circ W$ are different ways to ensure positivity of the weighted adjacency matrix (Wei et al., 2020).

It is acceptable to build on or reuse existing results, but proper citations need including, especially when the authors have acknowledged these works in the paper.

**Questions For Authors:**

The current theoretical analyses are conducted in the linear case. Could the authors further comment on the optimization behavior for the non-linear case, with respect to the results in Proposition 4 and 5?

**Relation To Broader Scientific Literature:**

The paper introduces a method to incorporate general ancestral constraints into differentiable structure learning frameworks. There are previous works addressing ancestral constraints yet with some limitations, namely Wang et al., which assumes knowledge of path lengths or Chen et al., (2016) which do not consider differentiable frameworks.

**Theoretical Claims:**

There is no issue with the correctness of the theoretical claims.

---

> ### Author Rebuttal · Authors · 2025-03-27
>
> Thank you for your careful review. We first address your major concerns regarding novelty and technical differences. We abbreviate differentiable structure learning for causal discovery as DCD.
>
> # Novelty over Partial Orders (Ban et al. (2024))
> You consider path existence as part of partial orders. In fact, path existence imposes a **stronger** constraint than partial ordering for DCD; it is partial ordering that forms a subset of path existence constraints. Consider this example:
>
> Suppose variables $A$ and $B$ are not reachable through any directed path in the true graph. Still, there exists a total ordering $\pi$ satisfying the true topological order where $A \prec B$. Here, the partial order constraint $A \prec B$ is correct (not contradicting ground truth), yet the path $A\leadsto B$ does not exist.
>
> Since the existence of path $A \leadsto B$ implies $A \prec B$, but not vice versa, path existence provides strictly stronger structural information. Additionally, optimizing DCD with path existence constraints introduces unique challenges (inequivalence and inherent gradient conflicts) absent in partial-order-based DCD.
>
> Therefore, DCD with ancestral (path existence) constraints represents a novel, significantly more challenging task than DCD with partial ordering.
>
> # Essential Technical Differences from Prior Work
> - Chen et al. (2016) consider discrete score-and-search methods, not encountering differentiability or the related issues described in Sec. 1.
> - Sun et al. (2023) focus exclusively on edge constraints.
> - Wang et al. (2024) assume the length of the path $(i,j)$ is known beforehand and formulate the loss as $\text{ReLU}(\epsilon - |W|^k_{i,j})$. Even ignoring this strong assumption, their formulation is theoretically incorrect and suffers from the same inequivalence issue as the naive loss $\text{ReLU}(\epsilon - \sum_k |W|^k_{i,j})$ discussed in Sec. 3.1. Moreover, they do not address the critical order-violation issue.
>
> Thus, it is rational to state that this work is the first to systematically (correctly) address DCD with ancestral constraints.
> # Evidence on Oder-Violation
> We now provide explicit evidence of the order-violation issue. Specifically, we report:
>
> - **@OrderViolation**: Number of constraints specifying paths that violate the recovered ordering (i.e., reversed paths).
> - **@FailedPaths**: Number of unsatisfied constraints.
> - **Data loss**, F1 score and SHD.
>
> We compare PE-NOTEARS-zero (zero initialization) with PE-NOTEARS (order-guided optimization) across varying path loss weights. Results (30 nodes, ER2 graph, linear SEM with Gaussian noise, 80% paths) are summarized below as **PE-NOTEARS / PE-NOTEARS-zero**.
>
> | PE loss weight  | 1           | 10          | 50          | 100          | 500          |
> | --------------- | ----------- | ----------- | ----------- | ------------ | ------------ |
> | F1              | 0.68 / 0.60 | 0.68 / 0.41 | 0.68 / 0.22 | 0.68 / 0.19  | 0.68 / 0.17  |
> | Data Loss       | 8.4 / 21.3  | 9.0 / 45.9  | 9.1 / 115.1 | 10.3 / 160.5 | 11.4 / 276.6 |
> | @OrderViolation | 0.0 / 0.1   | 0.0 / 0.8   | 0.0 / 3.7   | 0.0 / 6.7    | 0.0 / 19.0   |
> | @FailedPaths    | 0.3 / 9.3   | 0.2 / 10.7  | 0.5 / 27.2  | 1.2 / 22.2   | 0.7 / 24.2   |
>
> The order-violation issue from Example 1 and Proposition 5 occurs when path loss significantly impacts optimization. To simulate this, we vary the path-loss weight from 1 to 500. As the path-loss weight increases, @OrderViolation (the count of order-violating paths) grows substantially, dominating @FailedPaths, aligning with our identified issue. In this case, reducing path-loss impact with zero initialization leads to improved optimization. In contrast, PE-NOTEARS with order guidance consistently achieves stable and strong performance even under significant path-loss weights, aligning well with stable prior-based structure learning.
>
> These results clearly illustrate the order-violation issue arising from zero initialization and demonstrate the effectiveness of order guidance in resolving this issue.
>
> # Questions
>
> Propositions 4 and 5 also hold in the nonlinear setting:
>
> **Proposition 4:** Nonlinear models typically define edge weights $ W $ from parameters $ \theta $ through non-negative transformations, e.g., $ W_{i,j} = \sqrt{\sum_{q=1}^m\theta_{i,j,q}^2} $. In such cases, the absolute value of edge weights directly correlates with the magnitude of parameters. Since path existence constraints and acyclicity constraints push $ |W_{i,j}| $ in opposite directions, they consequently push parameters $ \theta_{i,j,q} $ in opposing directions. Thus, the gradient conflict identified in Proposition 4 persists in nonlinear settings.
>
> **Proposition 5:** In nonlinear models, regularization ensures parameters $ \theta $ remain bounded, implying edge weights $ W $ are also bounded. Thus, the argument presented for Proposition 5 directly applies.
>
> # Other Responses
> Due to the character limit, please see responses to Reviewer mwJt for this part.

---

> > ### Comment · Reviewer_MDE7 · 2025-04-04
> >
> > Thank you for your detailed responses.
> >
> > I understand that path existence constraint poses a challenge in its own right and appreciate the thorough analyses up to Section 4. I also understand the difference in the motivations of the two papers. However, when it comes to practical methodology, the technical formulation is indeed the same to me. Specifically, the final objective in Eq. (25) is the NOTEARS relaxation of the primal problem in Eq. (7) of Ban et al. The proposed objective ends up making use of partial/total ordering as done in Ban et al., since the knowledge of the ordering seems sufficient to deal with path absence/existence. This is what undermines the significance of the contribution to me. Furthermore, the improvement in performance compared to Li et al. seem insignificant to me.
> >
> > For the above reasons, I remain unconvinced in the practicality of the proposed method, and while I appreciate the authors' efforts, I maintain my evaluation.

---

> > > ### Author Response · Authors · 2025-04-04
> > >
> > > Thank you again for your detailed consideration. To clarify, **Equation (25) is not the final objective of our approach (not even as part of  the approach); rather, Equation (18) serves as our final objective function**, explicitly addressing path existence-based structure learning. This is stated on line 287, right column, of our paper. Here are our detailed responses relative to your concerns:
> > >
> > > 1. **Partial orders alone are insufficient to handle path existence constraints.** Partial orders can equivalently be viewed as constraints on path absence (Eq. 23 in our paper). Thus, while they prevent certain erroneous edges, they **do not actively constrain the existence of particular paths**. Practically, path existence-based differentiable structure learning **typically recovers more missing edges** compared to partial-order-based approaches, even at the risk of introducing a small number of erroneous edges (see our response to reviewer mwJt, point 4, and Figure 12 in Appendix D). This is particularly crucial in practical scenarios where **discovering previously unknown causal relationships is prioritized over strict correctness**.
> > >
> > > 2. **Major technical differences to Ban et al. (2024)**: **The only technical intersection** with partial-order-based methods (e.g., Ban et al., 2024) is our adoption of solutions from partial-order-based structure learning as initializations. However, this initialization is **solely to address the order-violation issue**, which we explicitly identify as **a unique challenge arising from path existence constraints**. **Our primary method and final objective function explicitly target path existence-based structure learning (Eq. 18).** The equation **(Eq. 25) you mentioned is not as part of our proposed methodological approach**. It is provided solely within Proposition 7 to clarify theoretical insights on why partial order-based solutions provide effective initialization.
> > >
> > > 3. **Comparison to Li et al. (2018)**: Li et al.'s method is a discrete search strategy explicitly designed to incorporate ancestral constraints by direct graph evaluation during the search. Given identical prior knowledge, it is **natural to observe comparable improvements** between discrete and differentiable structure learning methods, as **both correctly leverage prior knowledge.** Therefore, our observed performance improvements (with **differentiable structure learning consistently yielding stronger overall results**) align with expectations, highlighting the practical efficacy of our differentiable path existence-based approach.
> > >
> > > 4. **Contribution Clarification:** Our main contribution lies in enabling current differentiable structure learning methods to integrate ancestral constraints—the remaining significant structural constraints (apart from edge constraints and partial orders) not previously incorporated into differentiable structure learning. Technically, continuously optimizing with path existence constraints requires: **1) an equivalent characterization of path existence and 2) favorable optimization dynamics despite gradient conflicts between path existence and acyclicity**. Our paper identifies and addresses these issues explicitly. Practically, ancestral constraint-based differentiable structure learning combines the power of neural networks and GPU resources with the reliability of external knowledge-guided constraints to **efficiently and comprehensively recover authentic, previously unknown causal mechanisms**. Furthermore, this work provides new ideas applicable to broader AI research benefiting from NOTEARS-based causal analyses, such as computer vision [1], fault diagnosis[2], and multi-agent systems [3].
> > >
> > >    [1] Zhang, C., Jia, B., Edmonds, M., Zhu, S. C., & Zhu, Y. ACRE: Abstract causal reasoning beyond covariation. CVPR 2021.
> > >
> > >     [2] Dai, E., & Chen, J. Graph-Augmented Normalizing Flows for Anomaly Detection of Multiple Time Series. ICLR 2022.
> > >
> > >     [3] Ruan, J., Du, Y., et al. GCS: Graph-Based Coordination Strategy for Multi-Agent Reinforcement Learning. In Proceedings of the 21st International Conference on Autonomous Agents and Multiagent Systems.
> > >
> > > We appreciate your engagement with our responses and hope this clearly illustrates the practical and technical contributions of our work.

---

### Official Review · Reviewer_mwJt · 2025-03-01

**Overall Recommendation:** 4

**Summary:**

This paper introduces a framework for integrating ancestral constraints into differentiable structure learning of causal DAGs, addressing challenges in representing path existence and order violations. The authors propose a binary-masked characterization method and an order-guided optimization strategy to improve constraint adherence. Theoretical analysis and empirical evaluations on synthetic and real-world datasets demonstrate the method’s effectiveness.

**Claims And Evidence:**

All claims in the paper are well motivated and supported with theoretical analysis.

**Essential References Not Discussed:**

I’d encourage the authors to discuss “Scalable Differentiable Causal Discovery in the Presence of Latent Confounders with Skeleton Posterior” in the related work, as it offers an alternative approach to incorporating edge constraints into the optimization process without directly setting W_ij=0.

**Experimental Designs Or Analyses:**

Please refer to my comments above.

**Methods And Evaluation Criteria:**

1. The proposed method appears well-motivated and reasonable for integrating ancestral constraints into differentiable structure learning.
2. Would it be possible to compare with Wang et al.’s work?
3. What are the additional computational overheads (e.g., running time, number of iteration to converge) compared to standard NOTEARS?
4. Section 5.4: Why does adding more constraints sometimes lead to suboptimal performance? Some discussion on this trade-off would be helpful. For example, given a large dataset with over 200 nodes, would incorporating constraints overwhelm the optimization process and lead to a performance downgrade?
5. D.1. Time Complexity: The comparison appears somewhat vague. Can I interpret the actual running time as **PO-NOTEARS + PE-NOTEARS-path**? If so, the comparison between PE-NOTEARS-path and NOTEARS seems unfair, as PE-NOTEARS-path benefits from a well-optimized initial guess.

**Other Comments Or Suggestions:**

N/A

**Other Strengths And Weaknesses:**

Strengths
- The paper is well organized and the presentation is very clear.
- The method is clean and supported with theoretical analysis.

Weaknesses
- Some concerns in the evaluation. Please refer to my comments above.

**Questions For Authors:**

Please refer to my questions in "Methods And Evaluation Criteria."

**Relation To Broader Scientific Literature:**

The work is important because it provides a solid foundation for differentiable causal structure learning by enabling the incorporation of constraints beyond simple edge presence or absence.

**Theoretical Claims:**

I have not examined the proofs in detail.

---

> ### Author Rebuttal · Authors · 2025-03-29
>
> Thanks for your careful review. Here are our responses.
>
> # Responses
>
> 2. Wang et al. formulate the path existence loss as $\text{ReLU}(\epsilon - |W|^k_{i,j})$ (Eq. (22) in their paper), assuming a known path length $k$. Without this assumption, the formulation naturally generalizes to $\text{ReLU}(\epsilon - \sum_k |W|^k_{i,j})$, precisely matching the intuitive function $\bar{p}(W)$ introduced in Sec. 3.1. Thus, the ablation labeled *-intuitive* reflects Wang et al.'s method under unknown path length. We will clarify this explicitly in the revised manuscript.
>
> 3. See point 5.
>
> 4. Specifying path existence (PE) benefits the recovery of missing edges without need of exactly identifying them, but it also risks introducing extra edges along the path. The observed degradation in F1 (or increased SHD) when adding constraints arises because the number of newly introduced extra edges outweighs the number of correctly recovered missing edges. However, since our method begins optimization from a good solution obtained by integrating partial order (PO) constraints—where extra edges have already been mitigated—FDR of PE-NOTEARS consistently improves compared to NOTEARS.
>
>    For a clear insight, we report the results (including a new case of 200 nodes) of structure learning under PO and PE constraints as **PO-NOTEARS / PE-NOTEARS (/ NOTEARS)**. Settings are ER2 graphs, 20 samples, linear Gaussian SEM, and 80% paths.
>
>    | @Node        | 10          | 20          | 30          | 50          | 200                |
>    | ------------ | ----------- | ----------- | ----------- | ----------- | ------------------ |
>    | TPR          | 0.78 / 0.81 | 0.74 / 0.79 | 0.69 / 0.75 | 0.62 / 0.66 | 0.39 / 0.42 / 0.21 |
>    | FDR          | 0.09 / 0.13 | 0.13 / 0.19 | 0.15 / 0.19 | 0.22 / 0.26 | 0.20 / 0.24 / 0.53 |
>    | PathRecovery | 0.96 / 0.99 | 0.84 / 0.98 | 0.81 / 0.96 | 0.74 / 0.89 | 0.46 / 0.61 / 0.12 |
>
>    PE constraints improve the TPR over PO-NOTEARS and recover more missing causal paths, but they can also increase the FDR. Nevertheless, **PE-NOTEARS consistently outperforms NOTEARS**, including the case of 200 nodes. Importantly, explicitly enforcing path existence is crucial in scenarios where **uncovering previously unknown causal mechanisms** is prioritized over simply ensuring correctness for all identified causal edges.
> 5. You are correct. The reported runtimes for PO-NOTEARS and PE-NOTEARS-path reflect the two stages of PE-NOTEARS separately. We'll include the total PE-NOTEARS runtime in the revised manuscript. Iterations remain similar (~20–30) across NOTEARS, PO-NOTEARS, and PE-NOTEARS-path.
>    Additionally, we've reduced complexity to $O(d^3 \log d)$, as addressed in point 3 for reviewer tD5o.
> ## **Partial responses to Reviewer MDE7**
> ## Methods
> We assume you refer to Eq. (18) (path existence-based DCD) rather than Eq. (25) in step (2).
> 1. The main contribution is identifying this order-violation issue and effectively utilizing partial orders to mitigate it, rather than proposing a novel partial order-based DCD algorithm itself.
> 2. Proposition 7 states that arbitrary optima of NOTEARS correspond precisely to optima under certain total orderings, fundamentally differing from Proposition 3 by Ban et al. (which states convexity given a total ordering). This clarifies that total orderings fully characterize NOTEARS optima, highlighting that partial orders refine total orderings towards better solutions.
> ### Experiments
> 1. Score-and-search methods perform poorly on SEM-generated data due to scoring mismatch (e.g., BIC). Thus, we evaluate performance on real-world data (Sachs-500), reporting the F1 scores below under the same set of random paths.
>    | @Paths           | 0    | 25   | 50   | 75   |
>    | ---------------- | ---- | ---- | ---- | ---- |
>    | Li et al. (2018) | 0.33 | 0.43 | 0.41 | 0.48 |
>    | PENOTEARS        | 0.42 | 0.42 | 0.49 | 0.54 |
> 2. The method in Wang et al. (2024) corresponds exactly to our intuitive path existence formulation without known path-lengths (See point 2, responses to reviewer mwJt). Hence, the ablation labeled *-intuitive* reflects their approach, which will be noted.
> ## References
> We will introduce the relevant work by Deng et al. (2024) in the paper.
> ## Weaknesses
> We will explicitly cite Proposition 6 as Proposition 2 from Ban et al. (2024). However, Eq. (16) is independent of Ban et al. (2024): it defines a threshold-based binary mask indicating absent paths, following standard practice in DCD (originally from NOTEARS' acyclicity loss forbidding self-loops, see Sec. 2.3). Although Ban et al. (2024) also use path absence, their intent (partial orders) differs from ours, where we introduce $b(W)$ specifically to address the unnecessary issue of in path-existence characterization.
> ## Suggestions
> - We will add these citations in Sec. 3.1.
> - We apologize for the oversight regarding DAGMA codes and will correct it.
> - We will include a summary algorithm highlighting key contributions.

---

### Official Review · Reviewer_tD5o · 2025-03-14

**Overall Recommendation:** 3

**Summary:**

The paper addresses the challenge of integrating ancestral constraints into differentiable structure learning for causal directed acyclic graphs (DAGs). The key problem is how to incorporate prior knowledge about the existence or absence of paths between variables (ancestral constraints) into the learning process, which is typically formulated as a continuous optimization problem. The authors identify two main issues: the non-equivalence of relaxed characterizations for representing path existence and the order violations among paths during optimization.

To tackle these challenges, the paper proposes a **binary-masked characterization method** and an **order-guided optimization strategy**. The binary-masked method ensures an equivalent representation of path existence by selectively activating constraints based on whether a path already exists. The order-guided strategy enforces partial order constraints implied by path existence, ensuring that the optimization process avoids order-violating paths and converges to favorable optima.

### Key Contributions:
1. **First Systematic Approach**: This is the first paper to systematically address the integration of ancestral constraints into differentiable structure learning, allowing the use of abstract prior knowledge to guide the discovery of fine-grained causal mechanisms.

2. **Binary-Masked Characterization**: The authors propose a binary-masked continuous relaxation that accurately represents path existence, addressing the non-equivalence issue in previous relaxed characterizations.

3. **Order-Guided Optimization**: The paper introduces an optimization strategy that enforces partial order constraints derived from path existence, ensuring that the optimization process avoids suboptimal solutions caused by order violations.

### Methodology:
- **Path Existence Constraints**: The authors formulate the problem of path existence constraints using a continuous relaxation of the path existence condition. They show that previous relaxed characterizations fail to equivalently represent path existence and propose a binary-masked method to address this issue.

- **Order-Guided Optimization**: The optimization strategy begins by enforcing partial order constraints implied by path existence, deriving a DAG that satisfies all specified path orders. This order-consistent DAG is then used as the initial adjacency matrix to optimize the path existence-constrained problem.

### Theoretical and Empirical Validation:
- The paper provides theoretical justification for the correctness of the proposed approach, showing that the binary-masked characterization and order-guided optimization strategy effectively address the challenges of integrating ancestral constraints.

- Experimental evaluations on both synthetic and real-world datasets demonstrate the effectiveness of the proposed method. The results show that the approach outperforms baseline methods and ablation variants, achieving better adherence to path existence constraints and higher accuracy in recovering causal structures.

**Claims And Evidence:**

To the best of my knowledge, there is sufficient evidence to the claims.

**Essential References Not Discussed:**

N/A.

**Experimental Designs Or Analyses:**

The experimental design is good and sound.

**Methods And Evaluation Criteria:**

#### **Strengths:**

1. **Novel Contribution to Differentiable Structure Learning**:
   - The paper makes a significant contribution by addressing the integration of **ancestral constraints** (path existence constraints) into differentiable structure learning. This is a novel and important extension of existing methods, as it allows for the incorporation of abstract prior knowledge about causal relationships, which is often available in real-world applications.
   - The proposed **binary-masked characterization** and **order-guided optimization strategy** are innovative solutions to the challenges of representing path existence and avoiding order violations during optimization.

2. **Theoretical Justification**:
   - The paper provides a strong theoretical foundation for the proposed methods, including proofs and lemmas that justify the correctness of the binary-masked characterization and the order-guided optimization strategy. This theoretical rigor enhances the credibility of the approach.

3. **Empirical Validation**:
   - The authors conduct extensive experiments on both **synthetic and real-world datasets**, demonstrating the effectiveness of their approach. The results show that the proposed method outperforms baseline methods and ablation variants, particularly in terms of **path recovery rate** and **adherence to path existence constraints**.
   - The experiments also highlight the robustness of the method across different settings, including varying numbers of nodes, edge densities, and sample sizes.

4. **Generalizability**:
   - The proposed method is shown to be compatible with different backbone differentiable structure learning algorithms, such as **NOTEARS**, **DAGMA**, and **GOLEM**. This demonstrates the broad applicability of the approach across various frameworks.

5. **Addressing Order Violations**:
   - The **order-guided optimization strategy** is a key strength, as it effectively addresses the issue of order violations among paths, which can lead to suboptimal solutions. By enforcing partial order constraints, the method ensures that the optimization process converges to solutions that respect the specified path existence constraints.

---

#### **Weaknesses:**

1. **Non-Differentiability of Binary Mask**:
   - A significant drawback of the proposed method is that the **binary mask** $ b(\mathbf{W}) $ is **not differentiable**. This undermines the core idea of differentiable structure learning, as the optimization process relies on gradient-based methods. The non-differentiability of $ b(\mathbf{W}) $ could lead to instability during optimization and make the method less effective in practice.
   - The authors should consider alternative approaches to ensure differentiability, such as using smooth approximations of the binary mask or exploring other differentiable representations of path existence.

2. **Gradient Conflicts**:
   - The proposed constraints for ensuring path existence can lead to **gradient conflicts** with the acyclicity loss. This is a critical issue, as gradient conflicts can hinder the optimization process and result in suboptimal solutions.
   - A simpler and more effective solution would be to relax the problem by constraining the **absence of the reverse edge** $(j, i)$ instead of enforcing the existence of the path $(i, j)$. This can be done by setting $ p(W)_{ij} = 0 $, which avoids gradient conflicts and is easier to optimize. The authors should consider adding this approach as a baseline in their experiments.


3. **Computational Complexity**:
   - The proposed method involves computing $ p(W) = \sum_{k=1}^{d} |W|^k $, which has a time complexity of $ O(d^4) $. This is significantly higher than the $ O(d^3) $ complexity of standard acyclicity losses, making the method computationally expensive, especially for large graphs. One possible way is to use the approach in Zhang, Zhen, et al. "Truncated matrix power iteration for differentiable DAG learning." Advances in Neural Information Processing Systems 35 (2022): 18390-18402, which should be $O(d^3 \log d )$. A possibily further speed up can be find in Zhang et. al. Analytic DAG Constraints for Differentiable DAG Learning, ICLR 2025, which is $O(d^3)$.


   - While the authors suggest that GPU acceleration can mitigate this issue, the high computational cost remains a limitation, particularly for large-scale applications.

4. **Limited Discussion on Path Absence Constraints**:
   - The paper briefly mentions path absence constraints but does not explore them in depth. Path absence constraints are easier to enforce and align well with the acyclicity loss, as they do not cause gradient conflicts. The authors should consider expanding their discussion on path absence constraints and comparing their performance with path existence constraints.

---

#### **Suggestions for Improvement**:
1. **Address Non-Differentiability**:
   - The authors should explore differentiable approximations of the binary mask $ b(\mathbf{W}) $ or alternative representations of path existence that are fully differentiable. This would improve the stability and effectiveness of the optimization process.

2. **Incorporate Constraints without Gradient Conflicting**:
   - The authors should consider adding the simpler approach of constraining the absence of the reverse edge $(j, i)$ as a baseline. This approach avoids gradient conflicts and is easier to optimize, providing a useful comparison to the proposed method.

3. **Better Time Complexity**
  - The time complexity of the approach may be reduced to $O(d^3)$.

**Other Comments Or Suggestions:**

N/A

**Other Strengths And Weaknesses:**

N/A

**Questions For Authors:**

See above weakness and strengths.

**Relation To Broader Scientific Literature:**

I did not find a issue in the part.

**Theoretical Claims:**

I have checked the proof in detail and they should be correct.

---

> ### Author Rebuttal · Authors · 2025-03-27
>
> Thanks for your detailed review and thoughtful comments. Here are our responses.
>
> ### 1. Non-Differentiability of Binary Mask
>
> We provide empirical evidence that incorporating the binary masked path existence loss, $\bar{p}(W)\circ b(W)$, does not compromise optimization stability. To evaluate this, we define an edge as *unstable* if its weight lies near the threshold $\epsilon_0 = 0.3$ used to compute $b(W)$. Here, an edge is unstable when $|W_{i,j}| \in [2.9, 3.1]$.
>
> We then report the proportion of unstable edges relative to the total recovered edges for both PE-NOTEARS (which employs the binary mask) and the original NOTEARS method. Settings are: sample size $n=20$, ER2 graphs, Gaussian noise, and a path percentage $p=80$.
>
> | @Node    | NOTEARS         | PE-NOTEARS      |
> | ---- | --------------- | --------------- |
> | 10   | $3.4 \pm 2.7$%  | $5.5 \pm 4.7$%  |
> | 20   | $5.8 \pm 3.1$%  | $5.7 \pm 2.6$%  |
> | 30   | $5.5 \pm 1.8$%  | $5.4 \pm 1.5$%  |
> | 50   | $10.0 \pm 2.0$% | $10.0 \pm 2.1$% |
>
> The results show that the proportion of unstable edges in PE-NOTEARS is comparable to that in NOTEARS. If the non-differentiability of $b(W)$ were to introduce instability, we would expect a significantly higher number of edges with weights near the threshold, where the binary mask could abruptly flip between 0 and 1. However, the results indicate that this is not the case.
>
> Thus, by demonstrating that weights rarely fall into the narrow unstable interval, we confirm that the binary masked loss remains stable. The stable behavior of the underlying continuous function $\bar{p}(W)$ ensures that most updates occur in regions where $b(W)$ is constant, preserving smooth optimization dynamics. This evidence supports that the binary mask $b(W)$ does not adversely affect the stability of the optimization process.
>
> We also explored a *purely continuous formulation* for characterizing path existence, as detailed in Appendix B.2. This approach leverages the logical duality between path presence and absence, ensuring a strict correspondence with the existence of at least one path. However, the dual formulation involves computing a product over terms that grows exponentially with the number of nodes, which leads to numerical stability issues in practice.
>
> ### 2. Gradient Conflicts
>
> The gradient conflict arises inherently between path existence and acyclicity constraints, as illustrated in Proposition 4. When enforcing path existence, we push edge weights toward larger absolute values, whereas the acyclicity constraint forbids cycles, thereby pushing edge weights in the opposite direction. A loss in gradient consistent with acyclicity thus **cannot simultaneously enforce the existence of specific paths**.
>
> Your suggested formulation—the absence of reversed paths—actually represents partial ordering implied by path existence, which is a **weaker** condition that does not enforce explicit paths. Indeed, optimizing with partial-order constraints serves as the first part in our method to derive order-consistent initializations for paths, as detailed in Sec. 4.2.
>
> We also empirically compare partial-order-based and path-existence-based structure learning in Fig. 12 (Appendix D). Results demonstrate that partial-order-based methods satisfy fewer path-existence constraints than our explicit path-existence-based method. See point 4, responses to Reviewer mwJt for more details.
>
> ### 3. Computational Complexity
>
> Thank you for your suggestion. We implemented the accelerated matrix power (and gradient) computation algorithm—fast TMPI with $O(d^3\log d)$ complexity—proposed in *Truncated Matrix Power Iteration for Differentiable DAG Learning*. Below, we compare fast TMPI against the original direct matrix power operation under the setting: ER2 graph, linear SEM with Gaussian noise, and 80% prior paths. Results are presented as **Direct Matrix Power / fast TMPI** for PE-NOTEARS.
>
> | @Node    | 10          | 20           | 30            | 50            |
> | -------- | ----------- | ------------ | ------------- | ------------- |
> | Time (s) | 29.8 / 26.7 | 106.4 / 90.3 | 269.3 / 188.3 | 973.6 / 500.8 |
> | F1       | 0.87 / 0.88 | 0.77 / 0.77  | 0.68 / 0.68   | 0.61 / 0.61   |
>
> We observe that PE-NOTEARS with fast TMPI achieves a significant speedup on large-scale graphs while maintaining comparable performance. This confirms that fast TMPI integrates effectively into our approach, reducing time complexity from $O(d^4)$ to $O(d^3\log d)$, making it competitive with the $O(d^3)$ complexity of NOTEARS.
>
> ### 4. Discussion on Path Absence Constraints
>
> We will further discuss the connection between path absence constraints and partial orders to provide deeper insights. Additionally, we will present a direct comparison between path existence and path absence constraints (currently shown in separate figures) to improve clarity.

---

### Official Review · Reviewer_de6D · 2025-03-15

**Overall Recommendation:** 4

**Summary:**

The paper addresses the problem of incorporating ancestral (path) constraints into differentiable causal structure learning methods, specifically NOTEARS-style algorithms. The authors identify two key issues with existing differentiable formulations: a non-equivalence issue in previous continuous relaxations of path-existence constraints, and an order-violation issue where constraints might fail during gradient-based optimization. They propose a binary-masked characterization of the path-existence constraint that precisely captures path presence equivalently and an order-guided optimization strategy that initializes optimization using a DAG consistent with ancestral orders. The contributions consist of (i) a principled framework for differentiable causal discovery with arbitrary ancestral constraints, (ii) a new equivalently valid path constraint formulation, and (iii) an order-guided optimization strategy to satisfy constraints and achieve higher accuracy.

**Claims And Evidence:**

The claims made by the authors are generally well supported by theoretical proofs and fair experiments. In particular, the central theoretical claim--that the masked constraint formulation exactly characterizes path existence--is substantiated by clear lemmas and theorems (Theorem 1) with complete proofs provided in the appendix.

The empirical claims are also well supported by comprehensive experiments and ablation studies, demonstrating improvements in DAG accuracy (SHD and F1 scores) and better satisfaction of path constraints compared to naive methods and baseline NOTEARS. Ablations specifically isolate the effects of the proposed masked constraint and order-guided initialization, validating each independently.

**Essential References Not Discussed:**

In general the authors do a good job on referencing relevant literature. I think the authors could consider citing [1], which is an older foundational work on structural priors and partial orders, as well as [2], which is a more recent work also dealing with partial orders and notears-like differentiable structure learning.

[1] Heckerman, et al. "Learning Bayesian networks: The combination of knowledge and statistical data." Machine learning 1995.

[2] Deng et al. "Optimizing notears objectives via topological swaps". ICML 2023.

**Experimental Designs Or Analyses:**

The experimental design was sound and supports the authors' claims. Experiments varied constraints, initializations, and thresholds, thoroughly validating robustness and highlighting specific advantages. Minor points were noted, such as the absence of explicit testing for the scenario with incorrect priors (which could influence practical applicability). Nevertheless, I think the overall empirical evidence was compelling.

**Methods And Evaluation Criteria:**

I think the proposed methods and evaluation criteria are appropriate. The authors select established differentiable learners (NOTEARS, DAGMA, GOLEM) as baselines, conduct extensive evaluations using synthetic benchmarks varying graph sizes/types and amounts of prior knowledge, and utilize standard accuracy metrics (SHD, TPR, FDR, F1). Importantly, they explicitly measure a "path recovery rate", directly assessing constraint satisfaction--a reasonable metric for their problem setting.

**Other Comments Or Suggestions:**

No minor comments.

**Other Strengths And Weaknesses:**

Key strengths include the originality and theoretical rigor of the proposed masked path formulation, systematic empirical validation, robust experimental design, and clear presentation. The authors identified fundamental limitations of naive differentiable constraints and clearly justified their proposed solutions. The resulting method demonstrated practical effectiveness, especially in scenarios with limited data but high-quality expert knowledge.

Weaknesses primarily involve computational complexity ($O(d^4)$), limiting scalability to large graphs (though GPU acceleration partly mitigates this), reliance on threshold hyperparameters $(\epsilon,\epsilon_0)$, and unaddressed robustness to incorrect or noisy priors. These aspects, however, are openly acknowledged by the authors and represent reasonable future directions rather than critical flaws.

**Questions For Authors:**

1. How sensitive are your results to the choice of edge threshold $\epsilon_0$ used for binary masks, and how should practitioners select it in practice?

2. How would the algorithm behave if provided ancestral constraints are incorrect or contradictory? Can it gracefully handle noisy or uncertain priors?

3. Did the choice of DAG from partial-order initialization significantly affect the final results when multiple DAGs satisfy given partial orders? Should multiple initializations be tried?

4. Could the growing strength of the acyclicity penalty potentially override satisfaction of path constraints during optimization? How was this handled?

5. Do you have ideas or preliminary results on efficiently approximating or reducing computational complexity ($O(d^4)$) of the path existence formulation to scale beyond graphs of 50 nodes?

**Relation To Broader Scientific Literature:**

I think the work is well-situated within existing literature, clearly bridging classical approaches (eg. constraint-based or score-based methods utilizing ancestral constraints) with modern differentiable structure learning methods. It identifies clear limitations of previous differentiable methods and improves on recent works by offering a general, theoretically sound approach for arbitrary ancestral constraints.

**Theoretical Claims:**

I checked in fair depth the theoretical claims provided in the main text and skimmed the appendix. The authors' derivations seem sound.

---

> ### Author Rebuttal · Authors · 2025-03-29
>
> Thank you for your detailed review. We will discuss the relevant references you mentioned in the paper. Here are our responses to your questions.
>
> # Reply to Questions
>
> 1. The edge threshold $\epsilon_0$ is a standard parameter in differentiable structure learning, set to $0.3$ following the default in NOTEARS in the paper. To address your concern, we experiment with varying $\epsilon_0$ and report the results (**NOTEARS / PE-NOTEARS**) below. The settings are: 20 samples, 20 nodes, ER2 graphs, linear Gaussian SEM, and 80% paths.
>
>    | $\epsilon_0$ | 0.1         | 0.3         | 0.5         | 0.7         | 1.0         |
>    | ------------ | ----------- | ----------- | ----------- | ----------- | ----------- |
>    | FDR          | 0.58 / 0.57 | 0.43 / 0.29 | 0.27 / 0.19 | 0.16 / 0.11 | 0.11 / 0.02 |
>    | TPR          | 0.75 / 0.71 | 0.68 / 0.84 | 0.60 / 0.79 | 0.50 / 0.70 | 0.25 / 0.57 |
>    | PathRecovery | 0.90 / 0.91 | 0.83 / 1.00 | 0.70 / 0.98 | 0.52 / 0.88 | 0.21 / 0.72 |
>
>    When $\epsilon_0$ is set too small, noisy edge weights insignificant for causality frequently fall near the threshold, causing instability as the binary mask $b(W)$ repeatedly flips between 0 and 1. Conversely, when $\epsilon_0$ is too large, the path existence loss strongly conflicts with the data fit loss by forcing edge weights to exceed their true values. Thus, data fit can override the path recovery.
>
>    In practice, selecting $\epsilon_0$ can be guided by either the optimal path recovery rate (e.g., optimal around $\epsilon_0=0.3,0.5$ in this case) or the distribution of edge weights from NOTEARS (above the min region where edge weights are concentrated).
>
> 2. To evaluate error tolerance, we randomly introduce erroneous paths at a certain ratio (termed ErrorRatio) relative to correct ones (without creating cycles). Below, we report results comparing F1 scores and recovery rates of correct (@✅) and erroneous (@❌) paths for **NOTEARS / PE-NOTEARS**. The settings are: 20 nodes, ER2 graphs, 20 samples, linear Gaussian SEM, and 80% paths.
>
>    | ErrorRatio | 10%         | 20%         | 30%         | 40%         |
>    | ---------- | ----------- | ----------- | ----------- | ----------- |
>    | F1         | 0.62 / 0.68 | 0.62 / 0.70 | 0.62 / 0.52 | 0.62 / 0.32 |
>    | @✅         | 0.83 / 0.97 | 0.83 / 0.99 | 0.83 / 0.84 | 0.83 / 0.85 |
>    | @❌         | 0.20 / 0.50 | 0.23 / 0.61 | 0.20 / 0.40 | 0.17 / 0.38 |
>
>    When the ErrorRatio exceeds approximately 30%, PE-NOTEARS underperforms NOTEARS, exhibiting significantly reduced F1 recovery of correct paths. This result provides a practical estimate of the error tolerance of PE-NOTEARS.
>
> 3. Order-guided initialization is obtained by solving the partial order-based structure learning problem in Eq. (23), providing a stable initialization that typically does not require additional selection.
>
>    If you instead refer to an arbitrary initialization that satisfies partial orders without solving Eq. (23), we tested an intuitive initialization using the matrix $\epsilon_0 A$ (where $A$ is the path mask), denoted as PE-NOTEARS-A. We report results for **NOTEARS / PE-NOTEARS / PE-NOTEARS-A** below:
>
>    | @Node        | 20                 | 30                 | 50                 |
>    | ------------ | ------------------ | ------------------ | ------------------ |
>    | F1           | 0.62 / 0.77 / 0.67 | 0.52 / 0.68 / 0.59 | 0.45 / 0.61 / 0.55 |
>    | PathRecovery | 0.82 / 0.99 / 0.99 | 0.83 / 1.00 / 0.99 | 0.79 / 0.98 / 0.96 |
>
>    We observe that while the simpler partial order-consistent initialization (PE-NOTEARS-A) recovers most paths, it yields less optimal DAG structures compared to order-guided optimization. This is because solving Eq. (23) achieves a good optima simultaneously for partial orders and data fit, effectively mitigating conflicts with both acyclicity and data fit. In contrast, the partial-order-only initialization does not resolve conflicts with data fit, resulting in suboptimal performance.
>
> 4. Actually, our order-guided optimization effectively resolves the issue of acyclicity constraints overriding path existence. To demonstrate this, we experiment with large path existence loss weights, ensuring this loss dominates the data fit loss and leaves the main trade-off between path existence and acyclicity.
>
>    In the experiment, PE-NOTEARS (with order guidance) still achieves strong adherence to prior paths and maintains good structural metrics. In contrast, methods without order guidance suffer significantly from acyclicity constraints overriding path existence, resulting in numerous unsatisfied paths due to order violations.
>
>    Please refer to the related results in the section *Evidence on Order-Violation* of our responses to reviewer MDE7.
>
> 5. Following the suggestion of reviewer tD5o, we have reduced the complexity to $O(d^3 \log d)$. Please refer to point 3 in our responses to reviewer tD5o for detailed results.

---

### Decision · Program_Chairs · 2025-05-01

**Decision:**

Accept (poster)

**Comment:**

This paper studies the problem of incorporating ancestral constraints into differentiable causal structure learning, focusing on NOTEARS-style algorithms. The authors identify two key issues with existing differentiable formulations: a non-equivalence issue in previous continuous relaxations of path-existence constraints, and an order-violation issue where constraints might fail during gradient-based optimization. The authors' contributions consist of (i) a principled framework for differentiable causal discovery with arbitrary ancestral constraints, (ii) a new equivalently valid path constraint formulation, and (iii) an order-guided optimization strategy to satisfy constraints and achieve higher accuracy. The reviewers appreciate the originality and theoretical rigor of the proposed masked path formulation, systematic empirical validation, robust experimental design, and clear presentation.